# SAES-SVD: Self-Adaptive Suppression of Accumulated and Local Errors for SVD-based LLM Compression

**Xing Hu**[1*]    **Dawei Yang**[1*◇]    **Yuan Cheng**[1,2†]    **Zhixuan Chen**[1]    **Zukang Xu**[1]

[1]Houmo AI    [2]Nanjing University

## Abstract

The rapid growth in the parameter scale of large language models (LLMs) has created a high demand for efficient compression techniques. As a hardware-agnostic and highly compatible technique, low-rank compression has been widely adopted. However, existing methods typically compress each layer independently by minimizing per-layer reconstruction error, overlooking a critical limitation: the reconstruction error propagates and accumulates through the network, which leads to amplified global deviations from the full-precision baseline. To address this, we propose Self-Adaptive Error Suppression SVD (SAES-SVD), a LLMs compression framework that jointly optimizes intra-layer reconstruction and inter-layer error compensation. SAES-SVD is composed of two novel components: ❶ Cumulative Error-Aware Layer Compression (CEALC), which formulates the compression objective as a combination of local reconstruction and weighted cumulative error compensation. Based on it, we derive a closed-form low-rank solution relied on second-order activation statistics, which explicitly aligns each layer's output with its full-precision counterpart to compensate for accumulated errors. ❷ Adaptive Collaborative Error Suppression (ACES), which automatically adjusts the weighting coefficient to enhance the low-rank structure of the compression objective in CEALC. Specifically, the coefficient is optimized to maximize the ratio between the Frobenius norm of the compressed layer's output and that of the compression objective under a fixed rank, thus ensuring that the rank budget is utilized effectively. Extensive experiments across multiple LLM architectures and tasks show that, without fine-tuning or mixed-rank strategies, SAES-SVD consistently improves post-compression performance. For example, at a 0.2 compression ratio on LLaMA-7B, existing methods exhibit an average accuracy drop exceeding 0.05, whereas SAES-SVD restricts the drop to only 0.02. These improvements underscore the potential of SAES-SVD to effectively narrow the gap between compressed models and their full-precision counterparts, paving the way for more reliable compression of LLMs.

## 1 Introduction

Large language models (LLMs), including GPT Family (Achiam et al., 2023; Dettmers et al., 2022), LLaMA Family (Touvron et al., 2023a;b; Dubey et al., 2024), and Qwen Family (Bai et al., 2023; Team, 2024; Yang et al., 2025), have achieved state-of-the-art performance in tasks such as commonsense reasoning, document summarization, and few-shot learning. However, these advances come at a substantial cost: LLMs with billions of parameters demand enormous computational and memory resources, rendering them impractical for deployment on edge devices. To address this challenge, model compression has become a key research direction, with approaches such as quantization (Frantar et al., 2022; Hu et al., 2025), network pruning (Ashkboos et al., 2024), knowledge

---

◇Corresponding author

*Equal contribution

†This work was conducted during his internship at Houmo

distillation (Hinton et al., 2014), and low-rank decomposition (Yuan et al., 2023; Li et al., 2025). Among these, low-rank compression, typically implemented through Truncated Singular Value Decomposition (Eckart & Young, 1936), directly reduces both parameter count and computational cost

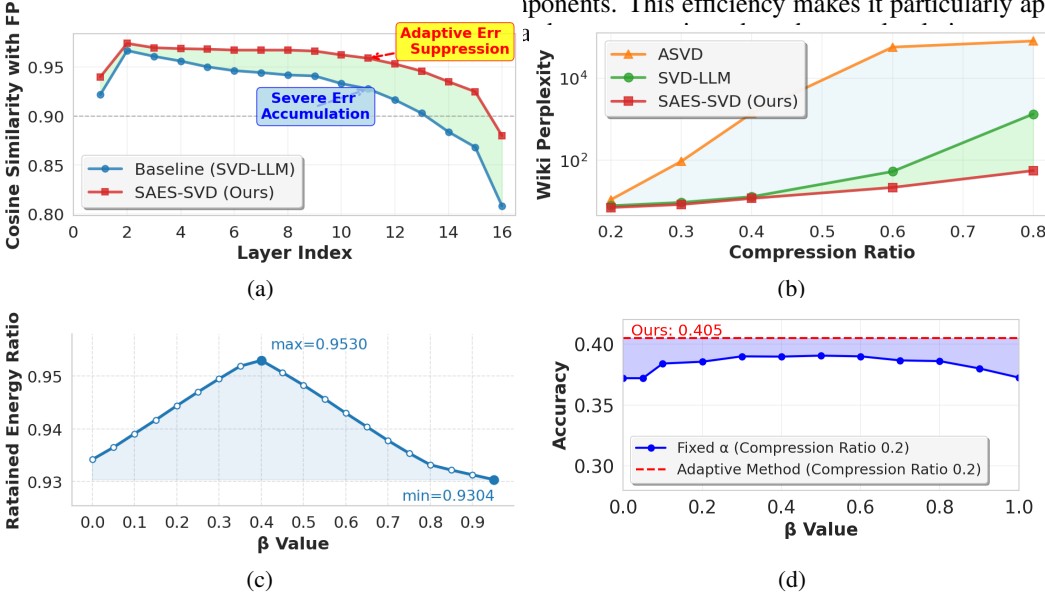

Figure 1: **(a)** Cumulative error analysis on LLaMA-3-8B, showing the cosine similarity of compressed outputs with the original reference across layers. SAES-SVD effectively mitigates error accumulation by adaptively suppressing upstream deviations. **(b)** Perplexity on WikiText2 under varying compression ratios. SAES-SVD consistently achieves lower perplexity than ASVD and SVD-LLM. **(c)** Retained Energy Ratio achieved by using different weighting coefficients for the cumulative error compensation term in with rank=128 compression for k-proj in the 14th Layer of LLaMA-3-8B. Here, for the sake of convenience, we use $\beta = \alpha/(1 + \alpha)$ as the horizontal axis. **(d)** Effect of fixed versus adaptive weighting coefficients for the cumulative error compensation on average accuracy across 7 zero-shot benchmarks.

Existing low-rank compression methods typically focus on layer-wise optimization. For example, methods such as ASVD (Yuan et al., 2023), SVD-LLM (Wang et al., 2024), and AdaSVD (Li et al., 2025) aim to reduce local reconstruction error within individual layers by introducing activation-aware scaling, applying whitening transformations, and performing adaptive iterative refinement, respectively. However, these approaches share a fundamental **limitation: they tend to minimize the reconstruction error for each layer independently, without considering how errors propagate and accumulate through the network.** In practice, reconstruction errors from early layers alter the input distribution of subsequent layers, causing these errors to compound and leading to an increasingly larger gap from the full-precision baseline. Applying SVD-LLM (Wang et al., 2024) to LLaMA2-7B further supports this observation. Although SVD-LLM achieves the theoretical minimum truncation error at each layer, the similarity to full-precision outputs, as shown in Figure 1a, decreases steadily from 0.97 to 0.79 across layers. This result indicates that minimizing per-layer reconstruction error alone does not guarantee end-to-end fidelity.

Building on these observations, we propose Self-Adaptive Error Suppression SVD (SAES-SVD), a low-rank compression framework that incorporates cumulative error compensation into the optimization objective. The core idea of SAES-SVD is to let each layer perceive accumulated compression errors from previous layers and adapt its compression strategy based on its sensitivity to those errors. In this way, conventional independent layer-wise optimization is transformed into a globally coordinated error-suppression mechanism. Specifically, SAES-SVD consists of two main components: ❶ **Cumulative Error-Aware Layer Compression (CEALC):** For each layer, we minimize not only the local reconstruction error but also enforce alignment with the corresponding full-precision reference. This alignment helps compensate for errors propagated from upstream compressed layers. The dual objective is formulated as a weighted Frobenius norm of reconstruction and alignment errors. From this objective, we derive a closed-form solution that effectively suppresses reconstruction error and compensates for cumulative error. Moreover, it achieves this

with only minimal additional cost, as the method relies solely on second-order statistics of activation deviations and a few extra matrix operations compared with vanilla SVD compression. ❷ **Adaptive Collaborative Error Suppression (ACES):** Since each layer has different sensitivity to compression ratio and cumulative error, using a fixed weighting coefficient (i.e., $\alpha_\ell$ in Equation 3) for cumulative error compensation often results in suboptimal solutions. As shown in Figure 1c, varying the coefficient across layers leads to different values of the Retained Energy Ratio (RER), defined as the proportion of the squared sum of the top-$k$ singular values to the total squared sum after SVD. RER serves as a measure of the low-rank structure of the compression objective, where a higher RER indicates stronger energy concentration in the leading singular values and thus greater compression efficiency. To address this issue, we introduce an adaptive weighting coefficient for each layer (i.e., $\beta_\ell^*$ in Equation 14). The coefficient is automatically tuned to maximize RER. By concentrating energy on the dominant singular values, this optimization yields compact and informative low-rank representations and enables energy-aware error compensation that improves compression efficiency under fixed rank budgets.

We conducted a comprehensive evaluation of SAES-SVD across diverse datasets, model architectures, and representative SVD-based baselines. The evaluation included language modeling, classification, and generation tasks. Results from multiple benchmarks, such as Figure 1, show that SAES-SVD effectively suppresses cumulative errors introduced during compression. Figure 1a indicates that cosine similarity with full-precision outputs remains consistently higher across layers. Figure 1b and Figure 1d further shows that SAES-SVD achieves much lower perplexity and higher accuracy under diverse compression ratios. The main contributions of this work are threefold:

- We identify that independent layer-wise compression leads to the amplification of cumulative errors. To address this, we propose CEALC, which moves beyond independent optimization and explicitly incorporates cross-layer error accumulation into the low-rank objective. Building on this formulation, we derive an efficient closed-form low-rank solution that depends only on second-order activation statistics.

- We introduce ACES, which use an adaptive coefficient $\alpha$ to control the weighting of cumulative error compensation. The value of $\alpha$ is adjusted to maximize the proportion of retained energy. This optimization enhances the low-rank structure of the compression objective and ensures that, under a fixed rank budget, the retained components preserve as much information as possible.

- CEALC and ACES together form the framework SAES-SVD. Experiments demonstrate that SAES-SVD is both effective and broadly applicable. At the 20% compression ratio, without requiring complex rank allocation or additional fine-tuning, it reduces the drop in zero-shot accuracy by 58% and lowers the perplexity gap by 52% compared with Dip-SVD. These improvements substantially narrow the gap between the compressed model and the full-precision baseline.

## 2 RELATED WORKS

**Large Language Model Compression Techniques.** The rapid growth of LLMs has spurred extensive research on compression methods that reduce model size and inference cost while preserving performance. Existing approaches can be broadly categorized into four classes: pruning, quantization (Nagel et al., 2021; Frantar et al., 2022; Hu et al., 2025; Lin et al., 2023; Hu et al., 2024), knowledge distillation (Hinton et al., 2014), and low-rank approximation (Yuan et al., 2023; Wang et al., 2024). Unstructured pruning (Ashkboos et al., 2024; Zhang et al., 2023) removes individual weights, but its irregular sparsity patterns often yield limited hardware benefits. Structured pruning (Ma et al., 2023; Dettmers et al., 2023) eliminates entire channels or blocks, improving efficiency but risking significant accuracy loss. Quantization (Frantar et al., 2022; Lin et al., 2023; Hu et al., 2025) reduces numerical precision, offering strong memory savings but typically restricted to 2–8 bits, which limits flexibility and may degrade performance under aggressive settings. Knowledge distillation transfers knowledge from a large teacher model to a smaller student (Liu et al., 2024), but requires costly retraining. In contrast, low-rank approximation, especially singular value decomposition (SVD), provides a hardware-agnostic, post-training alternative that can be combined with other methods, making it a particularly attractive solution for LLM compression.

**SVD-based Compression for LLMs.** SVD (Golub et al., 1987) reduces matrix sizes by truncating the smallest singular values. Low-rank approximation using SVD has been widely studied as a hardware-agnostic and efficient approach for compressing LLMs. Early works mainly focused on

minimizing the per-layer truncation loss. ASVD (Yuan et al., 2023) introduced a learnable scaling matrix to mitigate the loss of discarded singular values, but the improvement remained local and could not guarantee the theoretical minimum truncation loss across all layers. SVD-LLM (Wang et al., 2024; 2025b) addressed this limitation by incorporating a whitening matrix that normalizes input activations, thereby achieving theoretically minimal truncation error. AdaSVD (Li et al., 2025) introduces an alternating optimization scheme that iteratively updates low-rank matrices and compensation terms, thereby progressively reducing compression error. Several subsequent methods have incorporated importance weighting. For example, FW-SVD (Hsu et al., 2022) and GFW-SVD (Chekalina et al., 2025) leverage global importance indicators to guide single-layer compression, whereas DipSVD (Ding et al., 2025) jointly exploits both local and global importance to refine the weighting process. Although these techniques improve layer-level stability, they still rely on the assumption that layer errors are independent, leaving the problem of cumulative error unresolved.

## 3 PRELIMINARIES

For a weight matrix $W \in \mathbb{R}^{m \times n}$, its SVD is given by $W = U\Sigma V^\top$, where $U \in \mathbb{R}^{m \times m}$ is the left singular vector matrix, $V \in \mathbb{R}^{n \times n}$ is the right singular vector matrix, and $\Sigma \in \mathbb{R}^{m \times n}$ is a nonnegative diagonal matrix with entries arranged in descending order (with $\min(m, n)$ effective diagonal elements). The classical low-rank approximation problem can be formulated as

$$\min_{A \in \mathbb{R}^{m \times r}, \; B \in \mathbb{R}^{r \times n}} \|AB - W\|_F^2, \quad r \leq \min(m, n). \tag{1}$$

According to the Eckart–Young–Mirsky (Eckart & Young, 1936) theorem, the rank-$r$ optimal approximation of $W$ is $[W]_r = U_r \Sigma_r V_r^\top$. where $U_r \in \mathbb{R}^{m \times r}$ and $V_r \in \mathbb{R}^{n \times r}$ contain the top-$r$ left and right singular vectors of $W$. By choosing $A = U_r \Sigma_r^{1/2}$ and $B = \Sigma_r^{1/2} V_r^\top$, we obtain $AB = [W]_r$, thus achieving a rank-$r$ decomposition for compressing $W$. However, direct vanilla SVD compression ignores the statistical variations of layer input activations and often leads to significant accuracy degradation in LLM compression (Yuan et al., 2023; Wang et al., 2024). To address this issue, recent methods formulate LLM compression as an activation-aware, layer-wise independent optimization problem with the following objective:

$$\min_{A_\ell, B_\ell} \|A_\ell B_\ell X_\ell - W_\ell X_\ell\|_F^2, \tag{2}$$

here, $X_\ell$ denotes the input activations to the $\ell$-th layer, generated by the compressed upstream layers. In practice, compression in upstream layers introduces distributional shifts in these inputs. As depth increases, the gap between $X_\ell$ and its full-precision counterpart $X_\ell^f$ accumulates, eventually causing a substantial global deviation of the model outputs from the full-precision baseline.

## 4 METHOD

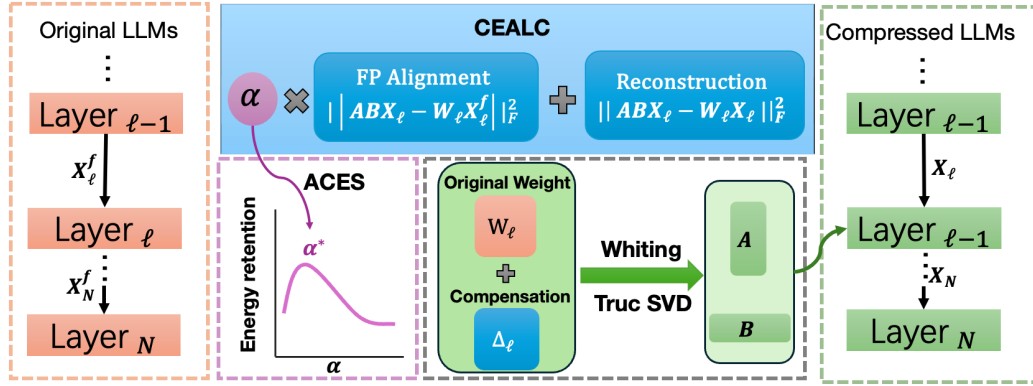

Figure 2: Overview of the proposed SAES-SVD framework.

We propose SAES-SVD, a unified low-rank compression framework that suppresses both per-layer's reconstruction errors and cumulative deviations accumulated from compressed upstream layers. As

illustrated in Figure 2, the framework comprises two key components. The first, Cumulative Error-Aware Layer Compression (CEALC), explicitly incorporates cumulative errors compensation into the compression objective, ensuring that each layer aligns closely with its full-precision counterpart. The second, Adaptive Collaborative Error Suppression (ACES), introduces an adaptive coefficient $\alpha$ to balance local reconstruction fidelity with alignment to the full-precision counterpart. The optimal $\alpha$ is chosen to maximize the proportion of retained energy within the truncated subspace.

### 4.1 CUMULATIVE ERROR- AWARE LAYER COMPRESSION

**Layer compression objective with cumulative error awareness**  Traditional SVD-based compression methods (Li et al., 2025; Wang et al., 2024) typically approximates the output of the current layer's weight matrix on its input $X_\ell$ by minimizing Equation 2. However, this approach neglects cumulative errors propagated from upstream compressed layers, which accumulate with increasing depth and amplify the global deviation from the full-precision (FP) baseline. To address this issue, we introduce an additional alignment term with respect to the FP reference output, leading to the following weighted objective:

$$\arg\min_{A_\ell, B_\ell} \underbrace{\|(A_\ell B_\ell - W_\ell)X_\ell\|_F^2}_{\text{intra-layer reconstruction error}} + \alpha_\ell \underbrace{\|A_\ell B_\ell X_\ell - W_\ell X_\ell^f\|_F^2}_{\text{FP reference alignment}} . \tag{3}$$

Here, $\alpha_\ell \geq 0$ controls the alignment strength. Let $T_\ell = W_\ell X_\ell$ be the original output under current inputs and $R_\ell = W_\ell X_\ell^f$ under FP inputs. Then Equation 3 is equivalent to

$$\|(A_\ell B_\ell X_\ell - T_\ell)\|_F^2 + \alpha_\ell\|(A_\ell B_\ell X_\ell - R_\ell)\|_F^2 = (1 + \alpha_\ell)\|A_\ell B_\ell X_\ell - Z_\ell\|_F^2 + C_\ell, \tag{4}$$

where the constant $C_\ell := \|T_\ell\|_F^2 + \alpha_\ell\|R_\ell\|_F^2 - (1+\alpha_\ell)\|Z_\ell\|_F^2$ is independent of $A_\ell, B_\ell$, and

$$Z_\ell = \frac{T_\ell + \alpha_\ell R_\ell}{1 + \alpha_\ell} = W_\ell \frac{X_\ell + \alpha_\ell X_\ell^f}{1 + \alpha_\ell}. \tag{5}$$

Therefore, Equation 3 is equivalent to

$$\min_{A_\ell, B_\ell} \|A_\ell B_\ell X_\ell - Z_\ell\|_F^2. \tag{6}$$

**Theorem 4.1.** *As illustrated by the detailed derivation in the Appendix A.2, let* $H_\ell = X_\ell X_\ell^\top \succ 0$ *and* $L_\ell = H_\ell^{-1/2}$. *Then, for any rank constraint* $r_\ell$,

$$\arg\min_{\text{rank}(U_\ell) \leq r_\ell} \|U_\ell X_\ell - Z_\ell\|_F^2 = \arg\min_{\text{rank}(V_\ell) \leq r_\ell} \|V_\ell - M_\ell L_\ell\|_F^2, \quad \text{with } V_\ell := U_\ell H_\ell^{1/2}. \tag{7}$$

Combined with Theorem 4.1, Equation 6 is equivalent to

$$\arg\min_{A,B} \|ABL^{-1} - ZX^\top L\|_F^2 \tag{8}$$

**Eliminating explicit inputs via second-order statistics.**  Storing raw activations $X_\ell$ and their FP counterparts $X_\ell^f$ requires prohibitive memory overhead. For instance, even with a moderate calibration set of 128 samples and sequence length 2048, the activation storage cost exceeds the parameter size by more than two orders of magnitude, making it infeasible for practical large-scale compression. To address this, we reformulate the objective in terms of compact second-order statistics. Specifically, we define the input covariance $H_\ell = X_\ell X_\ell^\top$ and the differential covariance $\Delta_\ell = (X_\ell^f - X_\ell)X_\ell^\top$, which capture the distributional characteristics and upstream-induced shifts. By substituting $X_\ell^f X_\ell^\top = H_\ell + \Delta_\ell$, we obtain

$$Z_\ell X_\ell^\top = \frac{W_\ell(X_\ell X_\ell^\top + \alpha_\ell X_\ell^f X_\ell^\top)}{1 + \alpha_\ell} = W_\ell\Big(H_\ell + \beta_\ell \Delta_\ell\Big), \qquad \beta_\ell := \frac{\alpha_\ell}{1 + \alpha_\ell} \in [0, 1). \tag{9}$$

Substituting Equation 8 into Equation 6, then the optimization objective becomes

$$\arg\min_{A_\ell, B_\ell} \|A_\ell B_\ell H_\ell^{1/2} - W_\ell(H_\ell + \beta\Delta_\ell)H_\ell^{-1/2}\|_F^2 \tag{10}$$

Hence, the target in Equation 6 depends only on $H_\ell$ and $\Delta_\ell$, which avoids explicit storage of $X_\ell$ and $X_\ell^f$. In practice, these two second-order statistics are collected in batches and layer by layer, similar to the strategy used in GPTQ, which greatly reduces space complexity.

**Closed-form via truncated SVD.** Letting $T_\ell = A_\ell B_\ell H_\ell^{1/2}$, and defining the target matrix $G_\ell = W_\ell(H_\ell + \beta\Delta_\ell)H_\ell^{-1/2}$, the Eckart–Young–Mirsky (Eckart & Young, 1936) theorem guarantees that the best rank-$r$ approximation of $G_\ell$ is obtained by truncated SVD:

$$T_l^* = [G_\ell]_{r_\ell} = \tilde{U}_{r_\ell} \Sigma_{r_\ell} \tilde{V}_{r_\ell}^\top, \tag{11}$$

where $\Sigma_{r_\ell}$ contains the top-$r_\ell$ singular values, and $\tilde{U}_{r_\ell}, \tilde{V}_{r_\ell}$ are the corresponding left and right singular vectors. Consequently, a closed-form solution for $A_\ell$ and $B_\ell$ is given by

$$A_\ell = \tilde{U}_{r_\ell} \Sigma_{r_\ell}^{1/2}, \qquad B_\ell = \Sigma_{r_\ell}^{1/2} \tilde{V}_{r_\ell}^\top L_\ell \tag{12}$$

with $L_\ell = H_\ell^{-1/2}$ precomputed from input statistics. At this point, we complete the derivation of the closed-form solution of CEALC. As shown in Equation 10, the approximation target consists of two weighting terms: $W_\ell H_\ell$ and $\beta\Delta_\ell$. The coefficient $\beta$ controls the strength of compensation for upstream cumulative errors when compressing the current layer $\ell$. After applying the whitening matrix $H^{-1/2}$, a truncated SVD yields the optimal theoretical solution.

## 4.2 ADAPTIVE COLLABORATIVE ERROR SUPPRESSION

**Motivation.** The formulation in Section 4.1 assumes that the alignment coefficient $\beta_\ell = \frac{\alpha_\ell}{1+\alpha_\ell}$ is fixed in advance. In practice, however, different layers have different sensitivities to accumulated errors from upstream layers. A single alignment strength can disrupt the low-rank structure of the original weights. As shown in Figure 1c, varying $\beta$ alters the low-rank characteristics of the compression objective in Equation 10. An inappropriate choice of $\beta_\ell$ scatters spectral energy into non-dominant components, which in turn reduces compression efficiency. To address this, we reinterpret Equation 10 as an approximation problem for the original weight matrix $W_\ell$. The key motivation is to permit moderate weight deviations, but only when such deviations help

- ① steers the compressed output toward the FP reference, thereby mitigating cumulative error; and
- ② maximizes the spectral energy preserved in the top-$r_\ell$ singular subspace, avoiding wasteful energy leakage into discarded components.

CEALC naturally satisfies condition ①, where the coefficient $\beta_\ell$ controls the strength of the alignment. Thus, our goal is to determine the optimal $\beta_\ell$ such that, under a rank budget $r_\ell$, the retained subspace preserves as much information as possible. Formally, we define

$$G_\ell(\beta_\ell) = S_\ell + \beta_\ell D_\ell, \quad \text{with} \quad \begin{cases} S_\ell = W_\ell H_\ell L_\ell, \\ D_\ell = W_\ell \Delta_\ell L_\ell, \\ \beta_\ell = \dfrac{\alpha_\ell}{1+\alpha_\ell} \in [0,1]. \end{cases} \tag{13}$$

The optimization target can then be expressed as maximizing the proportion of spectral energy captured by the top-$r_\ell$ singular values of $G_\ell(\beta_\ell)$:

$$\beta_\ell^\star = \arg\max_{\beta_\ell \in [0,1]} \frac{\sum_{i=1}^{r_\ell} \sigma_i^2(G_\ell(\beta_\ell))}{\sum_{i=1}^{\min(m_\ell, n_\ell)} \sigma_i^2(G_\ell(\beta_\ell))}, \tag{14}$$

where $\sigma_i(\cdot)$ denotes the $i$-th singular value in decending order after SVD. **However, singular values change with $\beta$, and each trial value would require a new SVD computation, making direct optimization intractable.**

**Theorem 4.2** (First-Order Approximation of Fixed Subspace(FS-FOA)). *Let $S_\ell \in \mathbb{R}^{m\times n}$ with singular value decomposition $S_\ell = U_\ell \Sigma_\ell V_\ell^\top$, and let $r_\ell \in \{1, \ldots, \min(m,n) - 1\}$ be the target rank. Denote by $U_{r_\ell} \in \mathbb{R}^{m\times r_\ell}$ and $V_{r_\ell} \in \mathbb{R}^{n\times r_\ell}$ the matrices formed by the top-$r_\ell$ left and right singular vectors of $S_\ell$, respectively. Define the orthogonal projection matrices*

$$P_L^\ell = I - U_{r_\ell} U_{r_\ell}^\top, \qquad P_R^\ell = I - V_{r_\ell} V_{r_\ell}^\top. \tag{15}$$

*For $G_\ell(\beta) = S_\ell + \beta D_\ell$, the relative error ratio can be approximated by*

$$\tilde{\rho}_\ell(\beta) = \frac{\|P_L^\ell G_\ell(\beta) P_R^\ell\|_F}{\|G_\ell(\beta)\|_F} = \frac{\|S_\ell^\perp + \beta D_\ell^\perp\|_F}{\|S_\ell + \beta D_\ell\|_F}, \tag{16}$$

*where $S_\ell^\perp = P_L^\ell S_\ell P_R^\ell$ and $D_\ell^\perp = P_L^\ell D_\ell P_R^\ell$ are the projections of $S_\ell$ and $D_\ell$ onto the orthogonal complements of the top-$r_\ell$ singular subspaces of $S_\ell$. Detailed proof can be referred to Appendix A.3.*

For computational tractability, we freeze the principal subspace at $\beta = 0$, leveraging Theorem 4.2. The resulting approximation ratio can be written as

$$\widetilde{\rho}(\beta) = \frac{\|S_\ell^\perp + \beta D_\ell^\perp\|_F^2}{\|S_\ell + \beta D_\ell\|_F^2} = \frac{a + 2b\,\beta + c\,\beta^2}{A + 2B\,\beta + C\,\beta^2}\,, \tag{17}$$

where

$$a = \|S_{\ell\perp}\|_F^2\,, \quad b = \langle S_{\ell\perp}, D_{\ell\perp} \rangle\,, \quad c = \|D_{\ell\perp}\|_F^2\,, \quad A = \|S_\ell\|_F^2\,, \quad B = \langle S_\ell, D_\ell \rangle\,, \quad C = \|D_\ell\|_F^2 \tag{18}$$

Setting the derivative of $\widetilde{\rho}(\beta)$ to zero yields a quadratic equation

$$\widetilde{p}(\beta) = (cB - bC)\,\beta^2 + (cA - aC)\,\beta + (bA - aB) = 0\,. \tag{19}$$

We evaluate all real roots within a feasible interval $[\beta\text{min}, \beta_{\max}]$ (e.g., $[0, 1]$), along with the endpoints and the minimizer $\beta^\star$ of $\widetilde{\rho}(\beta)$ is then selected. This adaptive selection minimizes the fraction of truncated energy, ensuring that the retained singular spectrum captures the maximum proportion of information. As illustrated in Figure 1d, the adaptive strategy consistently outperforms fixed-$\alpha$ settings across different compression ratios, highlighting both its robustness and effectiveness.

Through CEALC and ACES, we establish a closed-loop compression mechanism: each layer adaptively selects $\alpha_\ell$ to suppress newly introduced truncation error, while residual error is accumulated into second-order statistics and propagated forward. This allows subsequent layers to anticipate and compensate for upstream cumulative errors. Consequently, the model achieves output fidelity close to the baseline using only a single SVD approximation per layer. The detailed implementation can be found in Appendix A.11.

## 5   EXPERIMENTS.

**Models and Datasets.** To comprehensively evaluate the performance of SAES-SVD across diverse models and tasks, we conducted systematic experiments on mainstream open-source LLMs and standard language understanding and generation benchmarks, covering a wide range of parameter budgets and hardware settings. Unless otherwise specified, all results are obtained using a single-pass SVD per layer without task-specific fine-tuning after compression. For model selection, we considered representative architectures including LLaMA-1 (Touvron et al., 2023a), LLama-2 (Touvron et al., 2023b) and LLaMA-3 (LLaMA Team, AI @ Meta, 2024). Regarding evaluation tasks and datasets, we focused on natural language reasoning and understanding benchmarks. Specifically, we measured perplexity on the WikiText2 (Merity et al., 2016) and C4 (Raffel et al., 2020) dataset with a sequence length of 2048 to assess the language coherence and modeling capacity of the compressed models. In addition, we evaluated zero-shot accuracy on multiple datasets, including ARC-Challenge (Clark et al., 2018), ARC-Easy (Clark et al., 2018), HellaSwag (Zellers et al., 2019), MathQA (Amini et al., 2019), PIQA (Bisk et al., 2020), and WinoGrande (Sakaguchi et al., 2021), to comprehensively examine the generalizability of the compressed models on various tasks.

**Baselines.** In a unified calibration dataset and evaluation protocol, we compared our method with a range of well-established SVD-based low-rank compression approaches, including ASVD (Li et al., 2025), SVD-LLM (Wang et al., 2024), FW-SVD (Hsu et al., 2022), Dobi-SVD (Wang et al., 2025a), and AdaSVD (Li et al., 2025). Notably, SVD-LLM and Dobi-SVD rely on fine-tuning, while A-SVD, AdaSVD and Dobi-SVD employ mixed-rank strategies. In contrast, our proposed SAES-SVD does not require additional training nor auxiliary techniques, yet it consistently outperforms these baselines, highlighting its effectiveness and simplicity.

**Main Results.** We conducted a comprehensive evaluation of SAES-SVD for low-rank compression of LLMs and compared it with several SVD-based baselines across different compression ratios. Overall, SAES-SVD consistently achieved superior performance across tasks and model scales. Table 1 summarizes results on the representative LLaMA-7B. SAES-SVD reduced the perplexity gap by more than 35% compared with the strongest baseline and lowered the accuracy drop by 30–40%. At the more challenging 0.6 ratio, it halved the perplexity gap relative to Dobi-SVD and SVD-LLM, while keeping the accuracy drop below 0.03, compared with more than 0.05 for other methods. These results lead to two key observations: (i) SAES-SVD consistently reduces perplexity by a large margin—up to a $2\times$ improvement at aggressive compression levels—and (ii) it preserves or

Table 1: Perplexity and zero-shot evaluation of LLaMA-7B across seven benchmark datasets under varying compression ratios. The table compares SAES-SVD with competing SVD-based methods (ASVD* (Yuan et al., 2023), FWSVD (Hsu et al., 2022), SVD-LLM† (Wang et al., 2024), Dobi-SVD*† (Wang et al., 2025a), Dip-SVD*). Methods with fine-tuning are marked by †; mixed-rank strategies by *. Due to the closed-source implementation of Dip-SVD, we were unable to obtain its performance at the 0.4 compression ratio.

| Ratio | Method | Wiki2↓ | PTB↓ | C4↓ | Openb.↑ | ARC_e↑ | ARC_c↑ | WinoG.↑ | HellaS.↑ | PIQA↑ | MathQA↑ | Avg.↑ | Drop↓ |
|---|---|---|---|---|---|---|---|---|---|---|---|---|---|
| 0.0 | Baseline | 5.68 | 8.35 | 7.34 | 0.28 | 0.67 | 0.38 | 0.67 | 0.56 | 0.78 | 0.27 | 0.52 | 0.00 |
| 0.2 | FWSVD | 2e5 | 3e4 | 2e3 | 0.09 | 0.11 | 0.06 | 0.05 | 0.08 | 0.10 | 0.05 | 0.02 | 96% |
| | ASVD† | 11.14 | 16.55 | 15.93 | 0.25 | 0.53 | 0.27 | 0.64 | 0.41 | 0.68 | 0.24 | 0.43 | 16.7% |
| | SVD-LLM† | 7.94 | 16.22 | 15.84 | 0.22 | 0.58 | 0.29 | 0.63 | 0.43 | 0.69 | 0.24 | 0.44 | 14.7% |
| | Dobi-SVD*† | 8.54 | 14.83 | 10.01 | 0.26 | 0.59 | 0.31 | **0.66** | 0.44 | 0.70 | 0.23 | 0.46 | 10.8% |
| | Dip-SVD* | 7.95 | 15.60 | 14.07 | 0.27 | 0.63 | 0.33 | 0.64 | 0.45 | 0.71 | 0.24 | 0.47 | 9.2% |
| | **Ours** | **7.17** | **15.16** | **13.77** | **0.29** | **0.68** | **0.36** | 0.65 | **0.45** | **0.75** | **0.25** | **0.50** | **3.9%** |
| 0.4 | FWSVD | 2e4 | 1e4 | 1e4 | 0.06 | 0.05 | 0.02 | 0.02 | 0.00 | 0.05 | 0.03 | 0.02 | 96.6% |
| | ASVD† | 1e3 | 3e3 | 1e3 | 0.13 | 0.28 | 0.22 | 0.48 | 0.26 | 0.55 | 0.19 | 0.30 | 41.8% |
| | SVD-LLM† | 13.11 | 63.75 | 49.83 | 0.19 | 0.42 | 0.25 | 0.58 | 0.33 | 0.60 | 0.21 | 0.37 | 28.3% |
| | Dobi-SVD*† | 13.54 | 46.38 | 23.54 | 0.22 | 0.41 | 0.27 | 0.58 | 0.34 | 0.61 | 0.23 | 0.38 | 26.3% |
| | Dip-SVD* | 12.76 | 46.95 | 34.35 | 0.22 | 0.50 | **0.30** | 0.61 | 0.36 | 0.64 | 0.22 | 0.40 | 22.8% |
| | **Ours** | **10.42** | **45.13** | **32.79** | **0.23** | **0.50** | 0.29 | **0.62** | **0.36** | **0.65** | **0.23** | **0.41** | **21.1%** |
| 0.6 | FWSVD | 3e4 | 2e4 | 2e4 | 0.06 | 0.01 | 0.00 | 0.00 | 0.01 | 0.01 | 0.00 | 0.01 | 97.8% |
| | ASVD† | 6e4 | 4e4 | 4e5 | 0.12 | 0.26 | 0.21 | 0.49 | 0.26 | 0.53 | 0.18 | 0.29 | 43.8% |
| | SVD-LLM† | 53.74 | 4e2 | 3e2 | 0.14 | 0.28 | 0.22 | 0.50 | 0.27 | 0.55 | 0.21 | 0.31 | 39.9% |
| | Dobi-SVD*† | 46.18 | 2e2 | 2e2 | 0.15 | 0.31 | 0.20 | 0.52 | 0.28 | 0.54 | 0.22 | 0.32 | 38.0% |
| | **Ours** | **22.01** | **116.83** | **93.97** | **0.16** | **0.33** | **0.25** | **0.52** | **0.30** | **0.54** | **0.23** | **0.34** | **34.1%** |

Table 2: Zero-shot accuracy of LLaMA-13B under different compression ratios on seven benchmark datasets. Results are reported for SVD-LLM†, Dip-SVD*, and our proposed method. † denotes methods that rely on fine-tuning, while * indicates methods using mixed-rank strategies.

| Ratio | Method | Openb. | ARC_e | WinoG. | HellaS. | ARC_c | PIQA | MathQA | Average |
|---|---|---|---|---|---|---|---|---|---|
| 0.2 | SVD-LLM† | 0.302 | 0.683 | 0.684 | 0.470 | 0.356 | 0.725 | 0.265 | 0.498 |
| | Dip-SVD* | 0.306 | 0.681 | **0.692** | **0.490** | 0.369 | 0.734 | 0.258 | 0.504 |
| | **Ours** | **0.308** | **0.712** | 0.684 | 0.477 | **0.388** | **0.734** | **0.265** | **0.510** |
| 0.4 | SVD-LLM† | 0.222 | 0.521 | 0.639 | 0.355 | 0.248 | 0.637 | 0.228 | 0.407 |
| | Dip-SVD* | 0.230 | 0.548 | 0.644 | 0.402 | 0.283 | **0.661** | 0.233 | 0.429 |
| | **Ours** | **0.248** | **0.543** | **0.654** | **0.470** | **0.284** | 0.631 | **0.239** | **0.438** |

even improves accuracy, while other methods suffer notable degradation. Crucially, SAES-SVD outperforms not only methods requiring fine-tuning (e.g., SVD-LLM, Dobi-SVD) but also those relying on mixed-rank allocations (e.g., AdaSVD, Dobi-SVD,Dip-SVD). These findings, together with additional experiments shown in Table 5 and Table 6 in appendix, confirm that the gains of SAES-SVD generalize across architectures and model scales, underscoring both its robustness and broad applicability.

**Comparison on larger-scale models.** We further evaluated our method on larger architectures, including LLaMA-13B and LLaMA-30B, against current SOTA approaches. As shown in Table 2 and Figure 3, we conducted experiments across compression ratios ranging from 0.2 to 0.4. Our method consistently outperforms all baselines in terms of both average accuracy and perplexity. Notably, on LLaMA-30B, SAES-SVD reduces perplexity to 5.49, compared with 22.71 for ASVD and 5.63 for SVD-LLM( Note that Dip-SVD results are only available for 7B and 13B, since the original work does not report LLaMA-30B and its implementation is not publicly released), while achieving a 10% accuracy improvement over Dip-SVD. These results demonstrate that SAES-SVD remains highly effective at larger scales and delivers substantial gains in compressed model performance.

**Inference speed.** Low-rank approximation simultaneously decreases both computational complexity and parameter storage. This dual reduction substantially lowers memory footprint and compute requirements. We evaluated the inference speed of LLaMA3-8B under different compression ratios on a single NVIDIA A6000 GPU. As illustrated in Figure 4, our proposed SAES-SVD achieves consistent acceleration over the FP16 baseline, with speedups ranging from 1.29× to 3.79× as the compression ratio increases. These results demonstrate the practicality of SAES-SVD in accelerating inference while significantly reducing memory demands.

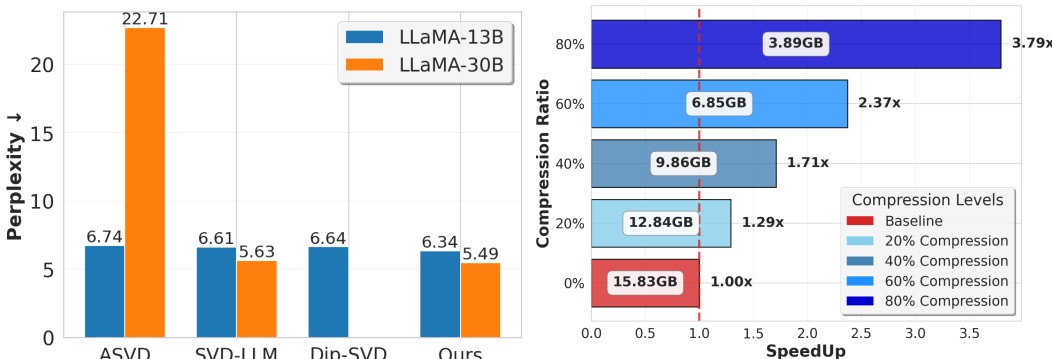

Figure 3: Wiki-PPL for two large-scale models, across different methods at 20% compression.

Figure 4: Memory usage and inference speedup of LLaMA-3-8B on varying compression ratios.

Table 3: Ablation study on the two components of our method. Avg Acc denotes the average accuracy over PIQA, ARC-e, HellaSwag and WinoGrande.

Table 4: Sensitivity analysis of the upper and lower bounds of the adaptive coefficient $\alpha$ in ACES, conducted on LLaMA2-7B at a 20% compression ratio.

| Model | CEALC | ACES | Wiki PPL↓ | Avg Acc↑ | $\alpha_{\min}$ | $\alpha_{\max}$ | Wiki PPL↓ | Avg Acc↑ |
|-------|-------|------|-----------|----------|------------------|------------------|-----------|----------|
| LLama2-7B | × | × | 9.34 | 58.66 | 0 | 1 | 8.96 | 58.47 |
| | ✓ | × | 7.66 | 62.02 | 0.1 | 0.9 | 7.88 | 62.81 |
| | ✓ | ✓ | 7.37 | 63.03 | 0.2 | 0.8 | 7.38 | 63.02 |
| LLama3-8B | × | × | 16.59 | 55.76 | 0.25 | 0.75 | 7.37 | 63.03 |
| | ✓ | × | 12.25 | 58.82 | 0.3 | 0.7 | 7.36 | 63.01 |
| | ✓ | ✓ | 11.48 | 60.18 | 0.4 | 0.6 | 7.63 | 62.87 |

**Ablation Experiments.** The proposed SAES-SVD framework consists of two complementary components: Cumulative Error-Aware Layer Compression (CEALC) and Adaptive Collaborative Error Suppression (ACES). CEALC addresses the limitation of prior methods that treat layer compression as independent processes; as shown in Figure 1a, it effectively mitigates cumulative compression errors. Building on this, ACES further enhances the low-rank properties of the compression objective shaped by CEALC, leading to superior performance under limited rank budgets, as illustrated in Figure 1d. The ablation results presented in Table 3 confirm these contributions: CEALC significantly improves both generative and zero-shot tasks, while ACES provides additional performance gains on top of CEALC.

We conducted a sensitivity analysis of the adaptive coefficient $\alpha$ by varying its lower and upper bounds. Experiments were performed on LLaMA-2 7B at a 20% compression ratio, with results summarized in Table 4. The analysis shows that performance is relatively stable within moderate ranges of $\alpha$. Specifically, when $\alpha_{\min} = 0.25$ and $\alpha_{\max} = 0.75$, the model achieves the best trade-off, with an average zero-shot accuracy of 63.03%. In contrast, setting the lower bound too high (e.g., $\alpha_{\min} > 0.8$) slightly degrades perplexity while reducing average accuracy significantly, indicating potential overfitting calibration dataset. These results validate our choice of setting the lower bound to 0.25 and the upper bound to 0.75 in practice.

## 6 CONCLUSION

In this work, we proposed SAES-SVD, a principled framework for low-rank compression of large language models that explicitly addresses both local reconstruction error and cross-layer accumulated error. By introducing Cumulative Error-Aware Layer Compression (CEALC) and Adaptive Collaborative Error Suppression (ACES), our method effectively mitigates error propagation while preserving critical low-rank structure. Extensive experiments across diverse models and tasks demonstrate that SAES-SVD consistently outperforms prior SVD-based methods in terms of perplexity and zero-shot accuracy, even under aggressive compression ratios, and does so without requiring fine-tuning or mixed-rank heuristics. These findings highlight the robustness, scalability, and practicality of SAES-SVD, providing a promising direction for deploying large language models efficiently in real-world scenarios.

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

## A    APPENDIX

### A.1    LLM DISCLAIMER

The authors hereby declare the role of large language model (LLM) tools in the preparation of this manuscript: LLMs were solely utilized to assist with text polishing (including refining sentence structure, optimizing lexical expression, and enhancing language fluency) and **writing optimization** of the paper's narrative content.

It is explicitly emphasized that all core components of this research, which determine the originality, scientific validity, and academic value of the work, were independently completed by the research team through manual efforts. These components include, but are not limited to:

- The formulation and development of the overall research framework, core ideas, and logical structure of the study;
- The design, coding, debugging, and validation of all algorithms and program codes involved in the research;
- The design of experimental protocols, collection and preprocessing of experimental data, execution of experiments, analysis and interpretation of experimental results, and verification of conclusions.

The use of LLM tools did not involve any participation in the conception of research content, generation of technical solutions, implementation of experimental processes, or derivation of research conclusions. All content of this paper adheres to academic integrity standards, and the research team assumes full responsibility for the scientificity, authenticity, and originality of the work.

A.2 THEORETICAL PROOF OF THEOREM 4.1 AND SUBSEQUENT TRUNCATED SVD

**Problem Formulation.**    Consider the optimization problem

$$\min_{A,B} J(A, B) = \|ABX - Z\|_F^2, \tag{20}$$

where the goal is to simplify the objective function $J(A, B)$ so that optimization with respect to $A$ and $B$ becomes more tractable. The key idea is to exploit the structure of the matrix $X$, particularly its row space.

**Assumption.**    To ensure that the following projection operations are well-defined, we assume that $X$ has full row rank, because in practice a small ridge parameter $\lambda$ will be introduced and $L_\ell = (H_\ell + \lambda I)^{-1/2}$. This assumption is reasonable since the activation matrix $X$ typically has far more columns than rows. If $X$ is a $p \times n$ matrix with $p \ll n$, then $\text{rank}(X) = p$, which guarantees that $XX^\top$ is invertible.

**Orthogonal Projection Operator.**    We define the projection matrix

$$P = X^\top (XX^\top)^{-1} X, \tag{21}$$

which is an $n \times n$ orthogonal projector mapping any row vector in $\mathbb{R}^n$ onto the row space of $X$. For any matrix $Y$ with $n$ columns, the product $YP$ projects each row of $Y$ onto the row space of $X$.

The matrix $P$ satisfies several important properties:

- **Symmetry:** $P^\top = P$, since $XX^\top$ is symmetric and so is its inverse.

- **Idempotence:** $P^2 = P$, which follows by direct calculation.

- **Projection property:** For any matrix of the form $CX$, right multiplication by $P$ leaves it unchanged, i.e., $(CX)P = CX$. In particular, $ABX$ belongs to this class.

**Decomposition of $Z$.**    We decompose $Z$ into components lying in the row space of $X$ and its orthogonal complement:

$$Z_P = ZP, \qquad Z_\perp = Z(I - P). \tag{22}$$

Clearly,

$$Z = ZP + Z(I - P) = Z_P + Z_\perp, \tag{23}$$

where $Z_P$ is the projection of $Z$ onto the row space of $X$, and $Z_\perp$ is the projection onto its orthogonal complement. Moreover, $Z_\perp P = 0$.

**Reformulated Objective.**    Using this decomposition, we expand the objective:

$$\begin{aligned} J(A, B) &= \|ABX - Z\|_F^2 \\ &= \|ABX - (Z_P + Z_\perp)\|_F^2 \\ &= \|(ABX - Z_P) - Z_\perp\|_F^2. \end{aligned} \tag{24}$$

Let $U = ABX - Z_P$ and $V = Z_\perp$. Since $U$ lies in the row space of $X$ and $V$ lies in its orthogonal complement, they are orthogonal, i.e., $\text{Tr}(U^\top V) = 0$. By the Pythagorean theorem for the Frobenius norm, we obtain

$$J(A, B) = \|ABX - Z_P\|_F^2 + \|Z_\perp\|_F^2. \tag{25}$$

By the premise, we start from

$$J(A, B) \;=\; \| ABX - ZX^\top (XX^\top)^{-1} X \|_F^2 \;=\; \| UX - ZP \|_F^2, \qquad where\, U = AB. \tag{26}$$

**Step 1: Projection reduction is tight up to a constant.** Decompose $Z$ into its components on the row space of $X$ and its orthogonal complement:

$$Z = ZP + Z(I - P) =: Z_P + Z_\perp, \qquad Z_\perp P = 0. \tag{27}$$

Since $UX$ lies in the row space of $X$, we have $(UX)P = UX$. Therefore,

$$\begin{aligned}
\|UX - Z\|_F^2 &= \|UX - (Z_P + Z_\perp)\|_F^2 = \|(UX - Z_P) - Z_\perp\|_F^2 \\
&= \|UX - Z_P\|_F^2 + \|Z_\perp\|_F^2 \quad \text{(orthogonality in the row/orthogonal subspaces).}
\end{aligned} \tag{28}$$

Thus minimizing $\|UX - ZP\|_F^2$ is equivalent to minimizing $\|UX - Z\|_F^2$, since they differ by the constant $\|Z_\perp\|_F^2$ independent of $U$. Hence equation 26 is a valid surrogate objective.

**Step 2: Whitening reformulation.** Let $M = ZX^\top \in \mathbb{R}^{m \times p}$. Expand the Frobenius norm using trace identities:

$$\begin{aligned}
\|UX - ZP\|_F^2 = \|UX - ZP\|_F^2 &= \|(UX - Z)P\|_F^2 = \mathrm{tr}\big((UX - Z)P(UX - Z)^\top\big) \\
&= \mathrm{tr}(UXX^\top U^\top) - 2\,\mathrm{tr}(UXPZ^\top) + \mathrm{tr}(ZPZ^\top) \\
&= \mathrm{tr}(UHU^\top) - 2\,\mathrm{tr}(UM^\top) + \mathrm{tr}(ZPZ^\top).
\end{aligned} \tag{29}$$

Because $H \succ 0$, its symmetric square roots $H^{\pm 1/2}$ exist and are invertible. Using

$$\mathrm{tr}(UHU^\top) = \|UH^{1/2}\|_F^2, \qquad \mathrm{tr}(UM^\top) = \langle UH^{1/2}, MH^{-1/2}\rangle_F, \tag{30}$$

we can complete the square:

$$\begin{aligned}
\|UX - ZP\|_F^2 &= \|UH^{1/2}\|_F^2 - 2\langle UH^{1/2}, MH^{-1/2}\rangle_F + \mathrm{tr}(ZPZ^\top) \\
&= \|UH^{1/2} - MH^{-1/2}\|_F^2 + \Big(\mathrm{tr}(ZPZ^\top) - \|MH^{-1/2}\|_F^2\Big).
\end{aligned} \tag{31}$$

The parenthetical term is independent of $U$. Therefore the minimizers of equation 26 are exactly the minimizers of

$$\min_{\mathrm{rank}(U) \le r} \big\|UH^{1/2} - MH^{-1/2}\big\|_F^2. \tag{32}$$

At this point, we have completed the proof of Theorem 4.1. Subsequently, we will perform low-rank approximation based on this.

**Step 3: Invertible change of variables and rank preservation.** Define the change of variables

$$V = UH^{1/2}, \qquad L = H^{-1/2}. \tag{33}$$

Since $H^{1/2}$ is invertible, $\mathrm{rank}(V) = \mathrm{rank}(U)$, so the rank constraint is preserved. The problem is thus equivalent to

$$\min_{\mathrm{rank}(V) \le r} \|V - ML\|_F^2. \tag{34}$$

This proves the desired whitening equivalence: minimizing $\|UX - ZP\|_F^2$ over rank-$r$ $U$ is equivalent to minimizing $\|V - ML\|_F^2$ over rank-$r$ $V$, with the bijection between minimizers given by $U^\star = V^\star H^{-1/2}$.

### A.3 THEORETICAL PROOF OF THEOREM 4.2: FIXED-SUBSPACE FIRST-ORDER APPROXIMATION FOR TRUNCATION RATIOS

**Setup.** Let $S \in \mathbb{R}^{m \times n}$ with singular value decomposition $S = U\Sigma V^\top$, and fix a target rank $r \in \{1, \ldots, \min(m, n) - 1\}$. Denote

$$U_r := U_{[:,1:r]}, \quad V_r := V_{[:,1:r]}, \qquad P_L := I - U_r U_r^\top, \quad P_R := I - V_r V_r^\top.$$

Given a direction $D \in \mathbb{R}^{m \times n}$ and a scalar $\beta \in [0, 1]$, define the affine path

$$G(\beta) := S + \beta D. \tag{35}$$

Let $\| \cdot \|_F$ be the Frobenius norm, $\| \cdot \|_2$ the spectral norm, and let $\sigma_1(\cdot) \geq \cdots$ denote singular values.

**True tail energy and ratio.** By the Eckart–Young–Mirsky theorem,

$$E(\beta) := \min_{\mathrm{rank}(Q) \leq r} \|G(\beta) - Q\|_F^2 = \sum_{i > r} \sigma_i\big(G(\beta)\big)^2. \tag{36}$$

We call $\sqrt{E(\beta)}$ the *tail (Frobenius) norm* of $G(\beta)$ at rank $r$. The corresponding *tail ratio* is

$$\rho(\beta) := \frac{\sqrt{E(\beta)}}{\|G(\beta)\|_F}. \tag{37}$$

**Fixed-subspace first-order approximation (FS-FOA).** Freeze the leading rank-$r$ subspaces at $\beta = 0$ (those of $S$), and approximate the tail by the projection onto the fixed orthogonal complement:

$$\widetilde{E}(\beta) := \big\| P_L\, G(\beta)\, P_R \big\|_F^2 = \big\| S_\perp + \beta D_\perp \big\|_F^2, \quad S_\perp := P_L S P_R, \; D_\perp := P_L D P_R. \tag{38}$$

The *FS-FOA tail ratio* is then

$$\widetilde{\rho}(\beta) := \frac{\|S_\perp + \beta D_\perp\|_F}{\|S + \beta D\|_F}. \tag{39}$$

**Assumption (spectral gap and small perturbation).** Let the spectral gap at $r$ be

$$\delta := \sigma_r(S) - \sigma_{r+1}(S) > 0, \tag{40}$$

and assume the relative perturbation is small:

$$\tau := \frac{\beta \|D\|_2}{\delta} \leq \tau_0, \qquad \text{for some sufficiently small } \tau_0 \in (0, 1). \tag{41}$$

**Theorem (FS-FOA is first-order accurate for the tail energy).** Under equation 40–equation 41, there exist absolute constants $C_1, C_2 > 0$ such that for all $\beta \in [0, 1]$,

$$E(\beta) = \|S_\perp + \beta D_\perp\|_F^2 + R_E(\beta), \qquad \big| R_E(\beta) \big| \leq C_1\, \tau^2 \Big( \|S\|_F^2 + \beta^2 \|D\|_F^2 \Big). \tag{42}$$

In particular, the discrepancy between the true tail energy and its FS-FOA surrogate is *second order* in $\tau$.

**Corollary (FS-FOA is first-order accurate for the tail ratio).** With the same assumptions, there exists $C' > 0$ such that

$$\big| \rho(\beta) - \widetilde{\rho}(\beta) \big| \leq C'\, \tau^2 = O\!\left( \frac{\beta^2 \|D\|_2^2}{\delta^2} \right). \tag{43}$$

Hence $\widetilde{\rho}(\beta)$ is a first-order accurate approximation to the true ratio $\rho(\beta)$.

**Proof sketch of equation 42.** Let $U_r(\beta), V_r(\beta)$ be the leading $r$-dimensional singular subspaces of $G(\beta)$ and $P_L(\beta) = I - U_r(\beta)U_r(\beta)^\top$, $P_R(\beta) = I - V_r(\beta)V_r(\beta)^\top$ their orthogonal complements. Standard Davis–Kahan/Wedin perturbation bounds imply

$$\|P_L(\beta) - P_L\|_2 \lesssim \tau, \qquad \|P_R(\beta) - P_R\|_2 \lesssim \tau. \tag{44}$$

Because $E(\beta) = \|P_L(\beta)G(\beta)P_R(\beta)\|_F^2$, write $P_L(\beta) = P_L + E_L$, $P_R(\beta) = P_R + E_R$ with $\|E_{L/R}\|_2 = O(\tau)$ and expand:

$$P_L(\beta)G(\beta)P_R(\beta) = (P_L + E_L)(S + \beta D)(P_R + E_R) = P_L(S + \beta D)P_R + \mathcal{E},$$

where the Frobenius norm of the remainder satisfies $\|\mathcal{E}\|_F = O(\tau\|G(\beta)\|_F)$ by the block structure induced by $(U_r, V_r)$. Squaring norms yields

$$E(\beta) = \|P_L G P_R\|_F^2 + O(\tau^2\|G(\beta)\|_F^2) = \|S_\perp + \beta D_\perp\|_F^2 + O(\tau^2(\|S\|_F^2 + \beta^2\|D\|_F^2)),$$

which is equation 42. The corollary equation 43 follows by dividing by $\|G(\beta)\|_F^2$ and using $\sqrt{1+x} = 1 + O(x)$ for small $x$.

**Closed form for the FS-FOA ratio and its stationary condition.** Both numerator and denominator of equation 39 are quadratic polynomials in $\beta$:

$$\|S_\perp + \beta D_\perp\|_F^2 = a + 2b\beta + c\beta^2, \tag{45}$$

$$\|S + \beta D\|_F^2 = A + 2B\beta + C\beta^2, \tag{46}$$

where

$$a := \|S_\perp\|_F^2, \quad b := \langle S_\perp, D_\perp \rangle, \quad c := \|D_\perp\|_F^2, \qquad A := \|S\|_F^2, \quad B := \langle S, D \rangle, \quad C := \|D\|_F^2. \tag{47}$$

Let

$$\phi(\beta) := \widetilde{\rho}(\beta) = \sqrt{\frac{a + 2b\beta + c\beta^2}{A + 2B\beta + C\beta^2}}. \tag{48}$$

Since the square root is monotone, $\arg\min \phi$ coincides with $\arg\min$ of the squared ratio. Differentiating and simplifying gives the *quadratic* stationary equation

$$(cB - bC)\beta^2 + (cA - aC)\beta + (bA - aB) = 0. \tag{49}$$

*Selection rule.* Solve equation 49, keep real roots in $[0,1]$, add the boundaries $\{0,1\}$, evaluate $\widetilde{\rho}(\beta)$ on this candidate set, and choose the minimizer $\beta^* \in [0,1]$. In applications with a trade-off parameter $\alpha \geq 0$, one uses the bijection

$$\beta = \frac{\alpha}{1 + \alpha} \quad \Longleftrightarrow \quad \alpha = \frac{1}{1 - \beta} - 1, \tag{50}$$

so that $\alpha^* = \frac{1}{\beta^*} - 1$.

**When is FS-FOA accurate.** Accuracy is controlled by $\tau = \beta\|D\|_2/\delta$. When $\tau \ll 1$ (e.g., $\tau \lesssim 0.5$), the error in equation 42–equation 43 is second order and the stationary points of $\widetilde{\rho}$ reliably approximate those of $\rho$. In near-degenerate cases (small spectral gap or large $D$), one may constrain $\beta \leq \beta_{\max} < 1$ or apply a mild shrinkage to the selected $\beta^*$; a one-step refinement that recomputes the leading subspaces at $G(\beta^*)$ further reduces the residual error.

**Interpretation.** The numerator in equation 39 measures the energy that would be *discarded* by rank-$r$ truncation (tail) when measured in the fixed orthogonal complement of the leading subspaces of $S$; the denominator measures the *total* energy. The quadratic form equation 49 encodes the balance between tail alignment $(a, b, c)$ and total alignment $(A, B, C)$, yielding a closed-form, one-dimensional selection of $\beta$ (hence $\alpha$) that minimizes the truncation ratio to first order.

## A.4 QUANTITATIVE RESULTS

Table 5: Performance of Compressed Llama-2-7B Across Benchmarks and Compression Ratios.

| Ratio | Method | WikiText-2 | PTB | C4 | ARC_e | WinoG. | HellaS. | PIQA | Average |
|---|---|---|---|---|---|---|---|---|---|
| 0 | Original | 5.68 | 8.35 | 7.34 | 74.62 | 69.22 | 76.00 | 79.11 | 68.85 |
| 0.4 | SVD | 39661.03 | 69493.00 | 56954.00 | 26.39 | 48.62 | 25.64 | 52.99 | 38.41 |
| | FWSVD | 8060.35 | 9684.10 | 7955.21 | 26.05 | 50.20 | 25.70 | 52.39 | 38.59 |
| | ASVD | 1609.32 | 7319.49 | 1271.85 | 26.81 | 49.49 | 25.83 | 53.81 | 38.99 |
| | SVD-LLM | 161.11 | 719.44 | 61.95 | 36.99 | 56.04 | 30.49 | 56.96 | 45.12 |
| | AdaSVD | 14.76 | 304.62 | 56.98 | 41.12 | 58.17 | 31.75 | 58.49 | 47.38 |
| | Ours | 11.35 | 217.20 | 40.57 | 43.27 | 57.77 | 32.14 | 58.92 | **48.03** |
| 0.5 | SVD | 53999.48 | 39207.00 | 58558.00 | 25.80 | 47.36 | 25.55 | 52.67 | 37.85 |
| | FWSVD | 8173.21 | 8615.71 | 8024.67 | 25.84 | 48.70 | 25.64 | 52.83 | 38.25 |
| | ASVD | 6977.57 | 15539.44 | 4785.15 | 25.13 | 49.17 | 25.48 | 52.94 | 38.18 |
| | SVD-LLM | 272.19 | 1772.91 | 129.66 | 31.65 | 51.14 | 28.38 | 54.57 | 41.44 |
| | AdaSVD | 25.58 | 593.14 | 113.84 | 34.18 | 54.06 | 28.88 | 55.50 | 43.16 |
| | Ours | 14.02 | 95.48 | 48.94 | 35.56 | 55.80 | 29.58 | 56.58 | **44.38** |
| 0.6 | SVD | 65186.67 | 79164.00 | 70381.00 | 24.49 | 51.85 | 25.40 | 53.16 | 38.73 |
| | FWSVD | 27213.30 | 24962.80 | 47284.87 | 25.38 | 48.46 | 25.61 | 51.96 | 37.85 |
| | ASVD | 10003.57 | 15530.19 | 9983.83 | 26.68 | 48.86 | 25.76 | 51.80 | 38.28 |
| | SVD-LLM | 89.90 | 2052.89 | 561.00 | 26.73 | 47.43 | 26.89 | 53.48 | 38.63 |
| | AdaSVD | 50.33 | 1216.95 | 239.18 | 28.20 | 51.22 | 27.36 | 52.83 | 39.90 |
| | Ours | 23.89 | 334.67 | 100.42 | 31.02 | 52.01 | 30.38 | 54.62 | **42.01** |

Table 6: PPL and accuracy of LLaMA-3-8B across compression ratios comparing SVD-LLM and Ours.

| Ratio | Method | WikiText-2 (PPL)↓ | Openb↑ | ARC_c↑ | ARC_e↑ | WinoG.↑ | HellaS.↑ | PIQA↑ | MathQA↑ | Mean↑ |
|---|---|---|---|---|---|---|---|---|---|---|
| 0.2 | SVD-LLM | 13.87 | 0.242 | 0.278 | 0.579 | 0.648 | 0.393 | 0.664 | 0.259 | 0.438 |
| | Ours | **11.49** | **0.252** | **0.284** | **0.593** | **0.658** | **0.393** | **0.671** | **0.268** | **0.446** |
| 0.4 | SVD-LLM | 80.46 | 0.134 | 0.193 | 0.325 | 0.523 | 0.274 | 0.548 | 0.208 | 0.315 |
| | Ours | **23.30** | **0.162** | **0.196** | **0.340** | **0.554** | **0.296** | **0.552** | **0.218** | **0.331** |
| 0.6 | SVD-LLM | 729.46 | 0.104 | 0.207 | 0.272 | 0.521 | 0.264 | 0.529 | 0.211 | 0.301 |
| | Ours | **63.09** | **0.136** | **0.219** | **0.285** | **0.534** | **0.278** | **0.535** | **0.225** | **0.316** |
| 0.8 | SVD-LLM | 6971.65 | 0.120 | **0.207** | 0.259 | 0.503 | 0.258 | 0.531 | 0.201 | 0.297 |
| | Ours | **181.84** | **0.140** | 0.203 | **0.281** | **0.520** | **0.259** | **0.539** | **0.208** | **0.307** |

We have supplemented detailed experimental results on different compression ratios for LLama-2 and LLama-3. As shown in Table 5 and Table 6, our method consistently achieves superior results across different compression ratios. Especially under extreme compression ratio conditions, such as Llama3-8B at a compression ratio of 0.8, our method outperforms the SVD-LLM method by nearly two orders of magnitude on the Perplexity metric.

## A.5 VISUALIZATION OF SELF-ADAPTATION BETA

To clarify the impact of different beta coefficients on the low-rank property of the optimization objective in Section 4.1, we visualized the compression of the k-proj Linear layer of the 14th layer of LLama3-8B. As shown in Figure 5, beta affects the low-rank property of the optimization objective. The difference between the most suitable beta and the worst beta within the range of [0,1] is significant. When beta = 0.95, as shown in the first subfigure, the proportion of discarded spectral energy is 7%, while when beta = 0.4, only 4.7% of the spectral energy is lost.

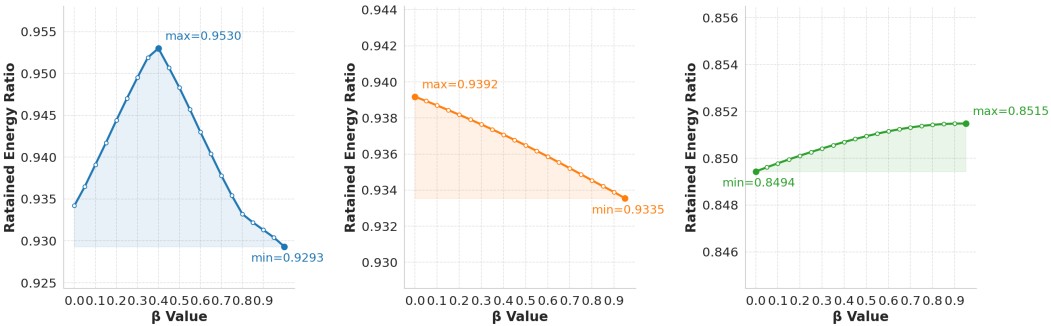

Figure 5: Using different beta coefficients for the k-proj of LLama3-8B at layer 14, layer 3 and layer 20 after the SVD in Equation 11, the proportion of retained energy at rank-1228.

We visualized in Figure 6 the Self-Adaptation beta coefficients of the odd k-proj layers of the LLama3-8B model under the condition of a compression ratio of 0.4. As can be observed from the figure, in the shallow layers of the network, beta tends to be small. As the number of layers deepens, the optimal beta continuously increases. In the last few layers, a downward trend appears.

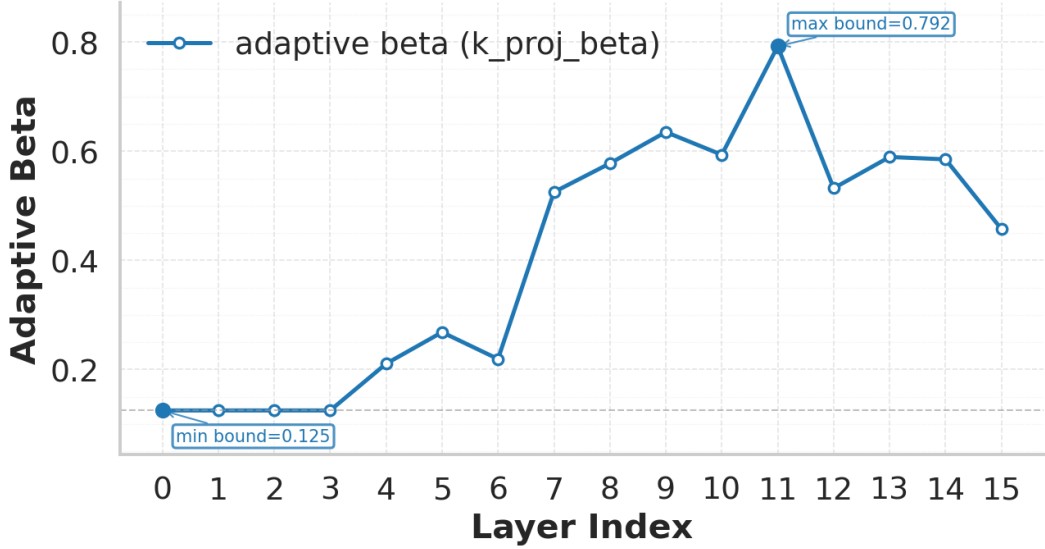

Figure 6: The visualization of the optimal beta values obtained from k-proj of odd-numbered layers on LLaMA2-7B.

A.6 PERFORMANCE ON MORE ARCHITECTURES AND MORE CHALLENGING TASKS

Table 7: Performance comparison of different compression methods on LLaMA3.1 and Qwen2.5 architectures at 0.2 compression rate across multiple benchmarks. Results are reported for 7 zero-shot tasks (ZeroShot[7]), long-context understanding (LongEval), code generation (HumanEval), and mathematical reasoning (GSM8K).

| Model | Method | ZeroShot[7] | LongEval | HumanEval | GSM8K |
|---|---|---|---|---|---|
| LLaMA3.1-8B | FP | 0.61 | 0.41 | 0.35 | 0.15 |
| | ASVD | 0.49 | 0.33 | 0.15 | 0.06 |
| | SVD-LLM | 0.57 | 0.37 | 0.27 | 0.10 |
| | **Ours** | **0.59** | **0.39** | **0.33** | **0.13** |
| LLaMA3.1-70B | FP | 0.67 | 0.49 | 0.55 | 0.37 |
| | ASVD | 0.54 | 0.39 | 0.32 | 0.26 |
| | SVD-LLM | 0.63 | 0.45 | 0.47 | 0.32 |
| | **Ours** | **0.66** | **0.48** | **0.52** | **0.35** |
| Qwen2.5-7B | FP | 0.60 | 0.58 | 0.66 | 0.71 |
| | ASVD | 0.50 | 0.49 | 0.41 | 0.42 |
| | SVD-LLM | 0.56 | 0.55 | 0.55 | 0.64 |
| | **Ours** | **0.58** | **0.57** | **0.63** | **0.69** |
| Qwen2.5-32B | FP | 0.65 | 0.79 | 0.55 | 0.72 |
| | ASVD | 0.52 | 0.61 | 0.37 | 0.44 |
| | SVD-LLM | 0.61 | 0.73 | 0.46 | 0.63 |
| | **Ours** | **0.64** | **0.76** | **0.51** | **0.68** |

Table7 reports the performance of LLaMA3.1 and Qwen2.5 families under a 0.2 compression ratio, comparing three representative low-rank decomposition approaches (ASVD, SVD-LLM, and our proposed SAES-SVD). Across all model scales and tasks, SAES-SVD consistently achieves the best reconstruction fidelity among compressed variants, demonstrating strong robustness in both general-purpose reasoning (ZeroShot[7]), long-context understanding (LongEval), code generation (HumanEval), and mathematical reasoning (GSM8K).

Notably, SAES-SVD narrows the performance gap to the FP baseline much more effectively than existing SVD-based methods. For example, on LLaMA3.1-8B, SAES-SVD retains 96.7% of FP performance on ZeroShot[7] and preserves near-lossness on LongEval. Similar trends hold for larger models: on LLaMA3.1-70B and Qwen2.5-32B, SAES-SVD significantly outperforms ASVD and SVD-LLM across all tasks, highlighting its superior scalability to high-capacity architectures. These results collectively demonstrate that SAES-SVD is a more faithful and stable low-rank approximation method, enabling high compression rates while minimizing degradation across diverse benchmarks.

A.7 THE EFFECT OF INTRODUCING THE MIXED-RANK CONFIGURATION

Table 8: Performance of SAES after incorporating the mixed-rank configuration derived from ASVD on the LLaMA-7B model. ZeroShot[7] denotes the averaged accuracy over seven zero-shot tasks consistent with the main evaluation protocol.

| Compression Ratio | Method | Wiki PPL | ZeroShot[7] |
|---|---|---|---|
| 0.2 | ASVD | 11.14 | 0.43 |
| | SAES-SVD | 7.17 | 0.50 |
| | SAES+mixrank | **6.19** | **0.51** |
| 0.4 | ASVD | 1.00E+03 | 0.30 |
| | SAES-SVD | 10.42 | 0.41 |
| | SAES+mixrank | **7.48** | **0.47** |

Table 8 summarizes the effect of introducing the mixed-rank configuration into the SAES framework for the LLaMA-7B model. The mixed-rank strategy is obtained by reusing the adaptive rank

Table 9: Perplexity comparison of SAES-SVD and three representative structured compression methods—LLM-Pruner, SliceGPT, and BlockPruner—on LLaMA-7B under various memory budgets. Lower perplexity indicates better language modeling quality.

| Memory | LLM-Pruner | SliceGPT | BlockPruner | SAES-SVD (Ours) |
|--------|-----------|----------|-------------|-----------------|
| 10GB | 9.88 | 8.78 | 9.40 | **7.17** |
| 9GB | 12.21 | 12.73 | 12.76 | **8.22** |
| 8GB | 18.94 | 16.39 | 19.78 | **8.96** |
| 7GB | 21.68 | 27.41 | 43.05 | **10.15** |

distribution discovered by ASVD, while the reconstruction and optimization procedure remains governed by our SAES architecture. Across both compression ratios(0.2 and 0.4), incorporating mixed rank consistently improves over standard SAES-SVD. At a 0.2 compression ratio, SAES+mixrank achieves the lowest Wiki perplexity(6.19) and the highest zero-shot accuracy (ZeroShot@7 = 0.51), outperforming both ASVD and SAES-SVD. A similar pattern holds at the more aggressive 0.4 compression ratio, where ASVD collapses severely in language modeling, but SAES+mixrank maintains stable performance with a substantial gain in AvgAcc@7(0.47). These results indicate that adaptive rank allocation and SAES's weight reconstruction are complementary: the mixed-rank configuration provides better structural flexibility, while SAES preserves semantic fidelity during compression. The combination yields a more robust low-rank approximation that scales effectively to higher compression budgets.

## A.8 COMPARISON WITH STRUCTURED PRUNING METHODS

Table 9 presents the perplexity of LLaMA-7B compressed using three widely adopted structured pruning and slicing techniques—LLM-Pruner, SliceGPT, and BlockPruner—together with our SAES-SVD method under progressively tighter memory budgets. Across all settings from 10GB down to 7GB, SAES-SVD consistently achieves the lowest perplexity, indicating superior retention of language modeling capability. When memory reduces to 10GB, SAES-SVD reaches a perplexity of 7.17, surpassing SliceGPT (8.78) and outperforming LLM-Pruner and BlockPruner by a clear margin. As compression becomes more aggressive, the gap further widens: at 8GB and 7GB budgets, structured pruning baselines degrade sharply—BlockPruner even collapsing to a perplexity above 40—while SAES-SVD maintains stable performance with perplexity 8.96 and 10.15, respectively. This demonstrates that SAES-SVD provides stronger parameter preservation under stringent memory constraints, largely due to its ability to reconstruct informative subspaces without relying on brittle structural sparsity assumptions. As a result, SAES-SVD serves as a more robust alternative to traditional structured compression pipelines, especially in low-memory deployment scenarios.

## A.9 COMBINATION CAPABILITY OF SAES-SVD AND GPTQ

Table 10: Comparison between pure GPTQ-3bit quantization and the equivalent 3bit configuration obtained by combining SAES-SVD with GPTQ on LLaMA2-7B and Qwen2.5-7B. WikiText2 perplexity and ZeroShot@7 represent language modeling quality and the averaged accuracy across seven zero-shot tasks, respectively.

| Model | Method | Wiki PPL | AvgAcc@7 |
|-------|--------|----------|----------|
| **LLaMA2-7B** | GPTQ (3bit) | 10.17 | 0.39 |
| | SAES+GPTQ (equal 3bit) | **7.64** | **0.44** |
| **Qwen2.5-7B** | GPTQ (3bit) | 12.29 | 0.51 |
| | SAES+GPTQ (equal 3bit) | **9.87** | **0.57** |

Table 10 compares the standard GPTQ 3-bit quantization with our equivalent 3-bit configuration that integrates SAES-SVD into the GPTQ pipeline. The evaluation covers both perplexity on WikiText2 and the averaged accuracy across seven zero-shot tasks for two representative models: LLaMA2-7B and Qwen2.5-7B. Across both architectures, the combined SAES+GPTQ strategy achieves a sub-

stantial improvement over pure GPTQ. For LLaMA2-7B, SAES+GPTQ reduces perplexity from 10.17 to 7.64 and increases AvgAcc@7 from 0.39 to 0.44. A similar trend is observed for Qwen2.5-7B, where perplexity improves from 12.29 to 9.87, accompanied by an accuracy gain from 0.51 to 0.57. These results demonstrate that SAES-SVD provides a complementary compression effect to GPTQ. By reconstructing a more informative low-rank subspace before quantization, SAES effectively smooths weight distributions and reduces quantization sensitivity, enabling GPTQ to operate at lower bit-widths without incurring substantial performance degradation. Consequently, the SAES+GPTQ pipeline offers a stronger alternative to pure 3-bit quantization while maintaining the same effective model size.

## A.10 THE THEORETICAL COMPLEXITY ANALYSIS AND COMPARISON OF COMPRESSION TIME

We provide a systematic evaluation from two perspectives—(1) **theoretical complexity analysis** and (2) **empirical runtime breakdown**—and compare our method (SAES-SVD) with the SVD-LLM baseline.

### A.10.1 THEORETICAL COMPLEXITY ANALYSIS

**Statistics Collection Phase.** For each layer with input dimension $d_{\text{in}}$, the unfolded calibration set contains $N$ tokens. We maintain only two second-order matrices:

$$H_\ell = X_\ell X_\ell^\top \in \mathbb{R}^{d_{\text{in}} \times d_{\text{in}}}, \quad \Delta_\ell = (X_\ell^f - X_\ell) X_\ell^\top \in \mathbb{R}^{d_{\text{in}} \times d_{\text{in}}}.$$

For each mini-batch with $b$ tokens, we perform two matrix multiplications with complexity $\mathcal{O}(d_{\text{in}}^2 b)$, and update the statistics via exponential moving averages. The total complexity over $L$ layers is:

$$\mathcal{O}\big(2L d_{\text{in}}^2 N\big),$$

which matches the order of GPTQ and other second-order methods, and is comparable to the statistics collection cost of SVD-LLM.

**Closed-form CEALC Solution for a Single Layer.** For a linear layer $W_\ell \in \mathbb{R}^{d_{\text{out}} \times d_{\text{in}}}$ with target rank $r_\ell$, the main computational costs of CEALC are:

- **Whitening matrix construction:** Compute $L_\ell = (H_\ell + \lambda I)^{-1/2}$ using a Cholesky-based procedure, with cost $\mathcal{O}(d_{\text{in}}^3)$. This cost is shared by all second-order compression methods.
- **Target matrix construction:** Build $G_\ell = W_\ell (H_\ell + \beta_\ell \Delta_\ell) L_\ell$, requiring several matrix multiplications with total complexity $\mathcal{O}(d_{\text{out}} d_{\text{in}}^2)$.
- **Truncated SVD:** Perform a rank-$r_\ell$ truncated SVD on $G_\ell$, with complexity $\mathcal{O}(d_{\text{out}} d_{\text{in}} r_\ell)$.

**ACES Adaptive Coefficient Selection.** During compression, ACES introduces additional computations:

- **Shared whitened matrices:** Construct

$$S_\ell = W_\ell H_\ell L_\ell, \quad D_\ell = W_\ell \Delta_\ell L_\ell,$$

each requiring $\mathcal{O}(d_{\text{out}} d_{\text{in}}^2)$.
- **Subspace extraction:** Perform a rank-$r_\ell$ truncated SVD on $S_\ell$ with complexity $\mathcal{O}(d_{\text{out}} d_{\text{in}} r_\ell)$.
- **Closed-form $\beta_\ell$ selection:** Using Theorem 4.2, compute a small number of Frobenius norms and inner products $(a, b, c, A, B, C)$ within the fixed subspace and solve a univariate quadratic equation in closed form. This step is negligible relative to the matrix operations.

Thus, compared with CEALC alone, ACES introduces only one extra truncated SVD per layer and inexpensive scalar operations.

**Fine-tuning Phase.** SVD-LLM requires approximately 3.5 hours of fine-tuning for LLaMA-7B, and even more for larger models. In contrast, **SAES-SVD requires no fine-tuning** and operates purely as a post-hoc compression method.

**Summary.** SAES-SVD and SVD-LLM share the same order of complexity in statistics collection and SVD-based factorization. However, by eliminating the fine-tuning phase, SAES-SVD substantially reduces the overall compression time.

### A.10.2 EMPIRICAL RUNTIME BREAKDOWN AND COMPARISON WITH THE BASELINE

Table 11: Comparison of SAES-SVD and SVD-LLM regarding statistics time, compression time, and fine-tuning time on LLaMA-7B.

| Method | Data Collection | Compression | Fine-tuning | Total | Reduction |
|---|---|---|---|---|---|
| SVD-LLM | 0.45h | 0.15h | 3.5h | 4.1h | – |
| SAES-SVD | 0.30h | 0.20h | 0 | **0.5h** | **-88%** |

We further provide in Table R3-1 an empirical breakdown of the compression time for SAES-SVD and SVD-LLM on LLaMA-7B, measured on an NVIDIA A6000 GPU. The results show:

- **More efficient statistics collection.** Although the theoretical complexity is similar, SAES-SVD uses a smaller calibration set (128 vs. 256 samples), reducing statistics collection time to about 67% of that of SVD-LLM.

- **Controlled compression overhead.** The SVD compression phase of SAES-SVD is slightly more expensive due to ACES, but remains within the same complexity order.

- **Elimination of fine-tuning.** SAES-SVD is a post-training method and does not rely on any fine-tuning, whereas SVD-LLM requires more than 3.5 hours of fine-tuning.

Overall, on LLaMA-7B, SAES-SVD achieves a total compression wall-clock time less than one-eighth of that of SVD-LLM, while obtaining better accuracy retention across multiple tasks. This combination of higher accuracy and lower end-to-end compression time highlights the practical efficiency of SAES-SVD for real-world deployment.

A.11 ALGORITHM

APPENDIX A. ALGORITHMIC DETAILS

**Algorithm 1 (CollectSecondOrderStats-Streaming).** This routine gathers the per-layer second-order statistics required by SAES-SVD without storing raw activations. For each layer $\ell$, it maintains a running estimate of the covariance $H_\ell = XX^\top$ and the cross-residual term $\Delta_\ell = (X^f - X)X^\top$. The update is fully streaming: existing statistics are reweighted by a factor $\gamma = n_\ell/(n_\ell + m)$ and the current mini-batch is scaled by $\sqrt{2/(n_\ell + m)}$ before accumulation, matching the numerics used in our implementation. Shapes are normalized to $(d_{\text{in}}, N)$ to keep the math consistent across layers. This design avoids caching $X_\ell$ or $X_\ell^f$ in memory, reduces I/O, and remains robust under variable batch sizes.

**Algorithm 2 (SAES-SVD for a single layer).** Given layer weights $W_\ell$ and the statistics $(H_\ell, \Delta_\ell)$, we form the whitener $L_\ell = (H_\ell + \lambda I)^{-1/2}$ and the whitened objective matrix $G_\ell(\beta) = W_\ell(H_\ell + \beta\Delta_\ell)L_\ell$. If no alignment strength is provided, $\beta$ is selected by Algorithm 3; otherwise we use the given $\alpha$ via $\beta = \alpha/(1+\alpha)$. A single rank-$r$ truncated SVD of $G_\ell(\beta)$ yields factors $A_\ell = \tilde{U}_\ell\Sigma_\ell^{1/2}$ and $B_\ell = \Sigma_\ell^{1/2}\tilde{V}_\ell^\top L_\ell$, which reconstruct a low-rank approximation of $W_\ell$ tailored to the cumulative-error–aware objective. The routine uses only one SVD per layer and a Cholesky-based whitener, making it both stable and efficient.

**Algorithm 3 (ACES_BetaSelect).** This procedure selects the adaptive alignment coefficient $\beta$ without repeated SVDs. We compute a *single* SVD of $S = WHL$ to obtain the rank-$r$ principal subspace, project $(S, D)$ onto the orthogonal complement to get $(S_\perp, D_\perp)$, and then optimize a first-order (fixed-subspace) surrogate. Two objectives are supported: (i) the *ratio* objective, which minimizes the approximate tail/total energy $\widetilde{\rho}(\beta) = \frac{a+2b\beta+c\beta^2}{A+2B\beta+C\beta^2}$; and (ii) the *energy* objective, which minimizes the tail energy $a + 2b\beta + c\beta^2$. Candidate $\beta$ values are obtained in closed form (stationary roots and interval endpoints), then filtered through guardrails: interval clipping $[\beta_{\min}, \beta_{\max}]$, a cap $\beta_{\text{cap}} < 1$ to prevent over-alignment, and an optional shrink factor $\rho \in (0, 1]$ for added stability. The final choice $\beta^\star$ is mapped back to $\alpha^\star = \beta^\star/(1 - \beta^\star)$ and passed to Algorithm 2.

**Complexity and implementation notes.** All three algorithms rely only on matrix multiplications, one Cholesky per layer for whitening, and one rank-$r$ truncated SVD per layer (randomized or Lanczos). No iterative backpropagation or fine-tuning is required. In practice, choosing a small ridge $\lambda$ ensures numerical stability when $H_\ell$ is ill-conditioned; if Cholesky fails, increase $\lambda$. The `ratio` objective in Algorithm 3 is recommended when cross-layer robustness is prioritized; the `energy` objective is a conservative alternative that further suppresses absolute tail energy.

---

**Algorithm 1:** CollectSecondOrderStats-Streaming (per-layer)

---

**Input:** Calibration dataset $\mathcal{D}$; layer set $\{\ell\}$; routines that expose per-layer inputs $X_\ell$ and FP references $X_\ell^f$

**Output:** Per-layer second-order stats $H_\ell \in \mathbb{R}^{d_{\text{in}} \times d_{\text{in}}}$, $\Delta_\ell \in \mathbb{R}^{d_{\text{in}} \times d_{\text{in}}}$

---

1 **foreach** *layer $\ell$* **do**
2    $H_\ell \leftarrow 0, \Delta_\ell \leftarrow 0, n_\ell \leftarrow 0$      `// running matrices and sample counter`
3 **foreach** *mini-batch $\mathcal{B} \subset \mathcal{D}$* **do**
    `// Forward once to cache both compressed-path inputs` $X_\ell$ `and`
      `FP references` $X_\ell^f$
4    run forward; collect $\{X_\ell, X_\ell^f\}_\ell$
5    **foreach** *layer $\ell$* **do**
      `// Flatten to 2D and transpose to` $(d_{\text{in}}, N)$
6      **if** $\text{ndim}(X_\ell) = 2$ **then**
7        $X \leftarrow X_\ell^\top$
8      **else if** $\text{ndim}(X_\ell) = 3$ **then**
9        $X \leftarrow \text{reshape}(X_\ell, -1, d_{\text{in}})^\top$
10      **else**
11        $X \leftarrow \text{flatten\_to\_2D}(X_\ell)^\top$
12      Do the same for $X_\ell^f$ to get $X^f$ with shape $(d_{\text{in}}, N)$
      `// Streaming reweighting (match code: scale old stats,`
       `add scaled current batch)`
13      $t \leftarrow n_\ell, \quad m \leftarrow N, \quad \gamma \leftarrow \frac{t}{t+m}$
14      $H_\ell \leftarrow \gamma H_\ell, \quad \Delta_\ell \leftarrow \gamma \Delta_\ell, \quad n_\ell \leftarrow t + m$
      `// Batch scaling by` $\sqrt{2/n_\ell}$ `(as in code)`
15      $s \leftarrow \sqrt{\frac{2}{n_\ell}}$
16      $\widetilde{X} \leftarrow s \cdot X, \quad \widetilde{X}^f \leftarrow s \cdot X^f$
      `// Update` $H_\ell = XX^\top$ `and` $\Delta_\ell = (X^f - X)X^\top$ `in the same metric`
17      $H_\ell \leftarrow H_\ell + \widetilde{X}\,\widetilde{X}^\top$
18      $dX \leftarrow \widetilde{X}^f - \widetilde{X}$
19      $\Delta_\ell \leftarrow \Delta_\ell + dX\,\widetilde{X}^\top$
20 **return** $\{(H_\ell, \Delta_\ell)\}_\ell$

---

**Algorithm 2:** SAES-SVD for layer $\ell$

---

**Input:** Weights $W_\ell \in \mathbb{R}^{d_{out} \times d_{in}}$; second-order statistics $H_\ell, \Delta_\ell$; target rank $r$; ridge $\lambda \geq 0$; (optional) $\alpha$ or $\beta$

**Output:** $A_\ell \in \mathbb{R}^{d_{out} \times r}$, $B_\ell \in \mathbb{R}^{r \times d_{in}}$ s.t. $W_\ell \approx A_\ell B_\ell$

1 $L_\ell \leftarrow (H_\ell + \lambda \mathbf{I})^{-1/2}$          `// Whitening Matrix`
2 **if** *$\alpha$ is given and $\beta$ is not* **then**
3    $\beta \leftarrow \alpha/(1 + \alpha)$
4 **if** *$\beta$ is not given* **then**
5    $\beta \leftarrow \text{ACES\_BetaSelect}(W_\ell, H_\ell, \Delta_\ell, r, \lambda)$
                                        `// Adaptive` $\beta$
6 $G_\ell \leftarrow W_\ell (H_\ell + \beta \Delta_\ell) L_\ell$       `// Whitened objective matrix`
7 $[\tilde{U}_\ell, \Sigma_\ell, \tilde{V}_\ell] \leftarrow \text{TruncatedSVD}(G_\ell, r)$     `// Trucated SVD Decomposition`
8 $A_\ell \leftarrow \tilde{U}_\ell \Sigma_\ell^{1/2}$
9 $B_\ell \leftarrow \Sigma_\ell^{1/2} \tilde{V}_\ell^\top L_\ell$
10 **return** $(A_\ell, B_\ell)$

---

---

**Algorithm 3:** ACES_BetaSelect (single-SVD, closed-form)

---

**Input:** $W \in \mathbb{R}^{m \times n}$,
$H = X X^\top$,
$\Delta = (X^f - X) X^\top$,
target rank $r$, damping $\lambda > 0$, interval $[\beta_{\min}, \beta_{\max}]$, cap $\beta_{\text{cap}} < 1$, shrink $\rho \in (0, 1]$, objective
$\in \{\texttt{ratio}, \texttt{energy}\}$
**Output:** $\beta^\star \in [\beta_{\min}, \beta_{\max}]$ and $\alpha^\star = \beta^\star / (1 - \beta^\star)$

**1** **Whitening and decomposition**:

**2** $L \leftarrow (H + \lambda I)^{-1/2}$ (Cholesky-based)

**3** $S \leftarrow W H L, \quad D \leftarrow W \Delta L$            // $G(\beta) = S + \beta D$

**4** $[U_r, \Sigma_r, V_r] \leftarrow \texttt{TopRSVD}(S, r)$          // only *one* SVD per layer

**5** **Projectors and FOA terms**:

**6** $P_L \leftarrow I - U_r U_r^\top, \quad P_R \leftarrow I - V_r V_r^\top$

**7** $S_\perp \leftarrow P_L S P_R, \quad D_\perp \leftarrow P_L D P_R$

**8** $a \leftarrow \|S_\perp\|_F^2, \ b \leftarrow \langle S_\perp, D_\perp \rangle, \ c \leftarrow \|D_\perp\|_F^2$

**9** $A \leftarrow \|S\|_F^2, \ B \leftarrow \langle S, D \rangle, \ C \leftarrow \|D\|_F^2$

**10** **Candidate generation (FOA)**:

**11** **if** *objective* = *ratio* **then**

**12**      $p(\beta) \leftarrow (cB - bC)\beta^2 + (cA - aC)\beta + (bA - aB)$     // stationary points of $\widetilde{\rho}(\beta)$

**13**      $\mathcal{C} \leftarrow \{\text{real roots of } p(\beta)\} \cup \{\beta_{\min}, \beta_{\max}\}$

**14**    $\beta_0 \leftarrow \text{clip}(-b/c, \ \beta_{\min}, \ \beta_{\max})$      // minimizes tail energy $a + 2b\beta + c\beta^2$

**15**    $\mathcal{C} \leftarrow \{\beta_0\}$

**16** **Selection with guardrails**:

**17** **foreach** $\beta \in \mathcal{C}$ **do**

**18**      $\beta \leftarrow \min\{\max\{\beta, \beta_{\min}\}, \beta_{\max}\}$

**19**      $\beta \leftarrow \rho \cdot \min\{\beta, \beta_{\text{cap}}\}$               // shrink & cap for stability

**20**      **if** *objective* = *ratio* **then**

**21**          $\text{score}(\beta) \leftarrow \dfrac{a + 2b\beta + c\beta^2}{A + 2B\beta + C\beta^2}$    // approx. tail/total ratio $\widetilde{\rho}(\beta)$

**22**      **else**

**23**          $\text{score}(\beta) \leftarrow a + 2b\beta + c\beta^2$                // tail energy

**24** **if** *objective* = *ratio* **then**

**25**    $\beta^\star \leftarrow \arg\min_{\beta \in \mathcal{C}} \text{score}(\beta)$

**26** **else**

**27**    $\beta^\star \leftarrow \arg\min_{\beta \in \mathcal{C}} \text{score}(\beta)$

**28** **Return**: $\beta^\star$ and $\alpha^\star = \beta^\star / (1 - \beta^\star)$

---

