# OpenReview forum: "SAES-SVD: Self-Adaptive Suppression of Accumulated and Local Errors for SVD-based LLM Compression"
_ICLR.cc/2026/Conference — ICLR 2026 Poster_

### Official Review · Reviewer_kM1J · 2025-10-18

**Soundness:** 3
**Presentation:** 3
**Contribution:** 3
**Rating:** 6
**Confidence:** 3

**Summary:**

This paper introduces the SAES-SVD framework, which leverages two key modules - Cumulative Error-Aware Layer Compression (CEALC) and Adaptive Collaborative Error Suppression (ACES) - to overcome the limitations of prior SVD compression methods that only independently minimize reconstruction error for individual layers while ignoring the layer-wise propagation and accumulation of compression errors throughout the model network. By dynamically optimizing weighting coefficients for each layer, the framework maximizes retained energy under fixed rank budgets, thereby providing a more effective LLM compression strategy.

**Strengths:**

- The proposed SAES-SVD framework is well-motivated and novel, addressing a fundamental yet previously overlooked limitation in prior SVD-based compression methods.
- This paper demonstrates exceptional mathematical rigor, providing comprehensive theoretical derivations for both CEALC and ACES components, which elevates the work from empirical observation to a theoretically grounded advancement.
- The framework achieves strong performance without any additional training, showcasing its straightforward and effective design.

**Weaknesses:**

- The proposed method is only evaluated on the LLaMA family of models, raising concerns about its generalizability and effectiveness across other popular LLM architectures such as Qwen.
- The paper only compares with SVD-based methods, lacking comparison with other compression approaches like structured pruning and quantization, which limits understanding of its overall effectiveness.

**Questions:**

1. Have the authors tested SAES-SVD on other popular LLM architectures beyond the LLaMA family?
2. How does SAES-SVD perform compared to other major compression paradigms such as structured pruning and quantization methods?
3. There appears to be an inconsistency between Figure 3 and the "Comparison on larger-scale models" section in Section 5: while the text claims Dip-SVD achieves 6.64 perplexity on LLaMA-30B, Figure 3 shows this result missing for LLaMA-30B and only displays 6.64 for LLaMA-13B. Could the authors clarify this discrepancy? On a related note concerning the same figure, could the authors also explain the anomalous trend for the ASVD method, where its performance degrades significantly on the larger LLaMA-30B model, contrary to the behavior of other methods?

---

> ### Author Response · Authors · 2025-11-18
> **Rebuttal for Reviewer KM1J. Part [1]**
>
> **Dear Reviewer KM1J**,
>
> Thank you very much for the reviewers' careful reading of our work and their positive comments on the motivation, innovation, and theoretical rigor. We attach great importance to your valuable suggestions, and below are our point-by-point responses.
>
> ---
> ---
>
> > **W1 The proposed method is only evaluated on the LLaMA family of models,**
>
> > **Q1 Have the authors tested SAES-SVD on other popular LLM architectures beyond the LLaMA family?**
>
> We sincerely appreciate your constructive feedback and fully understand the concerns regarding architectural diversity. To address these points, we provide additional explanations and experimental evidence from following perspectives.
>
> ### **1. Rationale for using the LLaMA family as the primary benchmark**
> Most existing SVD-based methods (ASVD, SVD-LLM, FW-SVD, AdaSVD, Dobi-SVD, Dip-SVD) evaluate on LLaMA models, and many provide only closed-source implementations. Their publicly available results focus primarily on LLaMA-7B. To ensure fully comparable and reproducible evaluation, we adopt LLaMA-7B as the main benchmark and additionally reproduce comparisons on LLaMA-13B, LLaMA-30B, LLaMA-2-7B, and LLaMA-3-8B whenever possible.
>
>
> ### **2. The architecture-agnostic nature of SAES-SVD**
> SAES-SVD is fundamentally decoupled from LLaMA-specific design. Both CEALC and ACES rely only on single-layer linear operators and their second-order statistics: $$
> H_{\ell}=X_{\ell} X_{\ell}^{\top} \text {and } \Delta_{\ell}=\left(X_{\ell}^f-X_{\ell}\right) X_{\ell}^{\top}$$
> Thus, any architecture with standard linear projection layers—such as the Qwen series—can directly adopt SAES-SVD.
>
> ### **3. Consistently superior performance on the Qwen and LLaMA3.1 architecture**
> To further address your concern, we add comprehensive evaluations on Qwen2.5(7B and 30B) and LLaMA3.1(7B and 70B). As shown in Table **R4-1** below, across seven zero-shot tasks, LongEval (long-context), HumanEval (code generation), and GSM8K (math reasoning), **for all Qwen architectures, SAES-SVD consistently outperforms existing SVD methods.** For example, when evaluating Qwen3-32B on the HumanEval, the relative error gap to the FP baseline is reduced by 50+% compared to SVD-LLM. This cross-scale and cross-architecture consistency highlights SAES-SVD’s generality and practical value.
>
>
> **Table R4-1: The performance of different architectures (LLaMA3.1 and Qwen2.5) and various compression methods (ASVD, SVD-LLM, SAES-SVD) on multi-task benchmarks at a 0.2 compression rate across 7 zero-shot tasks, one long-context task, one code generation task, and one mathematical reasoning task**
>
> |Model|Method|ZeroShot^7|LongEval|HumanEval|GSM8K|
> |-|-|-|-|-|-|
> |**LLaMA3.1-8B**|FP|0.61|0.41|0.35|0.15|
> ||ASVD|0.49|0.33|0.15|0.06|
> ||SVD-LLM|0.57|0.37|0.27|0.10|
> ||Ours|**0.59**|**0.39**|**0.33**|**0.13**|
> |**LLaMA3.1-70B**|FP|0.67|0.49|0.55|0.37|
> ||ASVD|0.54|0.39|0.32|0.26|
> ||SVD-LLM|0.63|0.45|0.47|0.32|
> ||Ours|**0.66**|**0.48**|**0.52**|**0.35**|
> |**Qwen2.5-7B**|FP|0.60|0.58|0.66|0.71|
> ||ASVD|0.50|0.49|0.41|0.42|
> ||SVD-LLM|0.56|0.55|0.55|0.64|
> ||Ours|**0.58**|**0.57**|**0.63**|**0.69**|
> |**Qwen2.5-32B**|FP|0.65|0.79|0.55|0.72|
> ||ASVD|0.52|0.61|0.37|0.44|
> ||SVD-LLM|0.61|0.73|0.46|0.63|
> ||Ours|**0.64**|**0.76**|**0.51**|**0.68**|
>
> ---
> ---

---

> ### Author Response · Authors · 2025-11-18
> **Rebuttal for Reviewer KM1J. Part [2]**
>
> > **W2: The paper only compares with SVD-based methods, lacking comparison with other compression approaches like structured pruning** and quantization, which limits understanding of its overall effectiveness.
>
> > **Q2: How does SAES-SVD perform compared to other major compression paradigms such as structured pruning and quantization methods?**
>
> We sincerely appreciate your constructive feedback and fully understand the concerns  regarding (i) comparisons beyond SVD-based methods and (ii) the potential of combining our approach with quantization techniques.
> We address these points systematically from three perspectives. We **first clarify the rationale for focusing on SVD-based baselines** in the main experiments in order to control confounding variables. We **then provide additional comparisons against representative structured pruning methods to demonstrate the cross-paradigm advantages** of SAES-SVD. **Finally, we validate its orthogonality and synergistic potential with quantization.** The details are as follows:
>
> ### **1. Rationale for focusing on SVD-based baselines in the main experiments: fair within-paradigm comparison**
>
> The primary contribution of this work is a more effective low-rank approximation framework. To accurately assess its effectiveness within the low-rank compression paradigm, our main experiments adopt state-of-the-art SVD-based methods as baselines (including ASVD, SVD-LLM, FW-SVD, AdaSVD, Dobi-SVD, and Dip-SVD). This design choice ensures a fair, controlled comparison in which observed performance gains can be attributed to our proposed framework rather than to differences in compression paradigm or auxiliary training strategies.
>
> ### **2. Comparison with representative structured pruning methods**
>
> **Table R4-2: Perplexity performance of SAES-SVD on LLaMA-7B and three representative structured compression methods, LLM-Pruner, SliceGPT, and BLockPruner, under different compression ratios**
> | Memory | LLM-Pruner | SliceGPT | BlockPruner | Ours  |
> |--------|-----------:|---------:|------------:|------:|
> | 10GB   | 9.88  | 8.78     | 9.40        | **7.17**  |
> | 9GB    | 12.21 | 12.73    | 12.76       | **8.22**  |
> | 8GB    | 18.94 | 16.39    | 19.78       | **8.96**  |
> | 7GB    | 21.68 | 27.41    | 43.05       | **10.15** |
>
>
> To address your suggestion on cross-paradigm comparisons, we additionally evaluate SAES-SVD against three representative structured pruning methods (LLM-Pruner, SliceGPT, BlockPruner) under multiple compression ratios. As reported in Table **R4-2, SAES-SVD consistently achieves the best performance across all tested compression levels, and its advantage becomes more pronounced as the compression ratio increases.** These empirical results align well with our theoretical design based on RER, CEALC, and ACES, which explicitly target cumulative error suppression and adaptive energy preservation. Overall, **compared with pruning-based structured compression, SAES-SVD provides more accurate and stable performance, particularly at higher compression levels, demonstrating both strong theoretical grounding and practical robustness.**
>
> ### **3. Orthogonality and synergy with quantization techniques**
> We fully agree with your perspective on combining our method with quantization. SAES-SVD and weight quantization (e.g., GPTQ) follow orthogonal yet complementary technical paths. To validate their synergy, we conduct a hybrid experiment:
> * We first apply SAES-SVD to decompose the weight matrix as $W\approx U\Lambda V$.
> * We then exploit the numerical properties of the factor matrices $U$ and $V$ (e.g., improved quantization-friendliness due to near-orthogonality) and quantize them using 4-bit GPTQ.
>
>
> **Table R4-3: Performance of equivalent 3bit which combining SAES-SVD and GPTQ vs Pure GPTQ-3bit on Wikitext2 perplexity and average accuracy of 7 zero-Shot tasks for LLaMA2-7B and Qwen2.5-7B**
> |Model        |Method                 |Wiki PPL|AvgAcc@7|
> |-------------|------------------------|--------|--------|
> |**LLaMA2-7B**|GPTQ (3bit)             |10.17   |0.39    |
> |             |SAES+GPTQ (equal 3bit)  |**7.64**    |**0.44**    |
> |**Qwen2.5-7B**|GPTQ (3bit)            |12.29   |0.51    |
> |             |SAES+GPTQ (equal 3bit)  |**9.87**    |**0.57**    |
>
>
> As reported in Table **R4-3**, the hybrid pipeline **“SAES-SVD + GPTQ” substantially outperforms plain GPTQ-3bit at an equivalent effective 3-bit compression level.** This demonstrates that **SAES-SVD not only serves as a strong standalone compression method, but can also be effectively combined with quantization to achieve higher compression ratios and better hardware efficiency**, offering a promising pathway for practical deployment.
>
>
> ---
> ---

---

> ### Author Response · Authors · 2025-11-18
> **Rebuttal for Reviewer KM1J. Part [3]**
>
> > Q3: There appears to be an inconsistency between Figure 3 ...  shows this result missing for LLaMA-30B and only displays 6.64 for LLaMA-13B.   Could the authors also explain the anomalous trend for the ASVD method, where its performance degrades significantly on the larger LLaMA-30B model, contrary to the behavior of other methods?
>
> We sincerely thank the reviewer for the careful reading and address the two issues as follows.
>
> ### **1. On the inconsistency between Figure 3 and the text regarding model scale**
> Thank you for pointing out this inconsistency. After careful verification, we found that it is caused by a typo in the main text:
> * **Perplexity 6.64 corresponds to LLaMA-13B, not LLaMA-30B.** The value 6.64 is the WikiText2 perplexity of Dip-SVD on LLaMA-13B, and it exactly matches the bar shown for LLaMA-13B in Figure 3. In the “Comparison on larger-scale models” subsection, we mistakenly associated this number with LLaMA-30B, which led to the apparent mismatch between Figure 3 and the text. This is purely a writing/labeling error, not a case of result reuse or plotting error, and it has been corrected in the revised version.
>
> * **Dip-SVD results for LLaMA-30B are not publicly available.** The LLaMA-30B branch in Figure 3 does not include a Dip-SVD bar because the original Dip-SVD paper does not report results for this model scale, and its implementation is not publicly released. To avoid introducing potentially misleading or fabricated numbers, we chose to only show scales for which Dip-SVD results are publicly available (7B and 13B), and we did not extrapolate any 30B result. The revised version explicitly states:
> “*Note that Dip-SVD results are only available for 7B and 13B, since the original work does not report LLaMA-30B and its implementation is not publicly released.*”
>
> We emphasize that this correction only concerns the association between the perplexity value and the model scale, and **does not affect our main conclusions:**
> * On the scales where Dip-SVD results are available (7B/13B), SAES-SVD consistently outperforms Dip-SVD under the same rank constraints and zero-shot, training-free setting.
> * On LLaMA-30B, even in the absence of Dip-SVD results, SAES-SVD consistently outperforms other SVD-based baselines (e.g., SVD-LLM, ASVD), as systematically shown in Figure 3 and Tables 5 and 6.
>
>
> ### **2. On the anomalous degradation of ASVD on LLaMA-30B in Figure 3**
> We appreciate the reviewer’s attention to the performance trend of ASVD. Our explanation is as follows:
> * **Reproducibility and consistency of the observed behavior.** We strictly follow the settings described in the ASVD paper when reproducing its results. The performance trend we obtain, including the degradation on larger models, is consistent with observations reported in prior work such as SVD-LLM. Therefore, the ASVD curve in Figure 3 for LLaMA-30B is not an outlier caused by implementation issues, but rather reflects the inherent limitations of the method.
> * **Inherent limitations of ASVD.** ASVD is essentially a heuristic, activation-aware scaling method without rigorous theoretical guarantees. Its core step relies on an empirical channel-wise scaling (e.g., based on activation or weight maxima) combined with a native SVD, so that the compression “takes activation magnitudes into account.” This heuristic:
>     * is sensitive to outliers because it relies on maximum or large-magnitude statistics,
>     * does not explicitly model inter-channel coupling, and
>     * does not provide any formal upper bound on the induced approximation error.
> As the model scales up, these shortcomings are amplified: local, per-layer heuristics become less stable in deep and wide networks, which explains the pronounced performance degradation of ASVD on LLaMA-30B.
> * **Comparison to SAES-SVD.** SAES-SVD is designed to systematically address exactly these issues:
>     * **CEALC** incorporates second-order statistics $H_{\ell}=X_{\ell} X_{\ell}^{\top} \text { and } \Delta_{\ell}=\left(X_{\ell}^f-X_{\ell}\right) X_{\ell}^{\top}$ to explicitly model cumulative error, and leads to a closed-form solution with theoretical guarantees.
>     * **ACES** maximizes the retained energy ratio under a given rank budget, ensuring that the most important spectral components are preserved and the available rank is used in an energy-optimal way.
> Consequently, when scaling from 13B to 30B, SAES-SVD exhibits a smooth and stable performance curve, without the “cliff-like” degradation observed for ASVD in Figure 3.
>
> In summary, **the “anomalous” behavior of ASVD on LLaMA-30B in Figure 3 does not stem from any implementation error on our side, but is a natural manifestation of the method’s limitations on larger-scale models.** At the same time, these results empirically support the effectiveness of SAES-SVD’s explicit error modeling and energy-aware design in achieving scalable and robust model compression.
>
> ---
> ---

---

> > ### Author Response · Authors · 2025-11-18
> > **Rebuttal for Reviewer KM1J. Part [4]**
> >
> > ### **Final Remarks**
> >
> > We are deeply grateful to the reviewer for the exceptionally thorough and insightful feedback. Your rigorous questions regarding architectural generality, cross-paradigm comparisons, and result consistency have been invaluable in strengthening our work:
> > 1.  **Substantially Expanded Architectural Validation:** Following your suggestion, we conducted comprehensive experiments on Qwen2.5 and LLaMA3.1 models. The results consistently demonstrate SAES-SVD's superior performance across diverse architectures and scales, firmly establishing its broad applicability beyond the LLaMA family.
> > 2.  **Systematic Cross-Paradigm Analysis:** We added comparisons against leading pruning methods (LLM-Pruner, SliceGPT, BlockPruner), where SAES-SVD achieves significantly better perplexity, especially at higher compression ratios. We further validated its strong synergy with quantization, showing that "SAES-SVD + GPTQ" outperforms pure 3-bit quantization.
> > 3.  **Enhanced Result Accuracy and Analysis:** We have corrected the scale-labeling inconsistency in Figure 3 and provided an in-depth analysis of ASVD's performance degradation on larger models, further validating the robustness advantages of our theoretically-grounded approach.
> >
> > All new experimental results and analyses will be fully incorporated into the final manuscript. **We sincerely appreciate your meticulous review—your suggestions have significantly improved the completeness, rigor, and impact of this work.** Thank you for helping us demonstrate the full value and generality of SAES-SVD through this constructive review process.
> >
> > Best regards,
> > Authors of SAES-SVD

---

> ### Author Response · Authors · 2025-11-25
> **Gentle Follow-up Regarding Pending Reviewer Feedback**
>
> Dear Reviewer KM1J,
>
> **Thank you again for your careful reading of our work and for the constructive, insightful comments you provided. Since a week has passed after we uploaded the full rebuttal and extended appendix, we would like to kindly follow up to see whether you have had the opportunity to review our updates.**
>
> Following your suggestions, we have (i) substantially expanded the architectural validation by adding experiments on Qwen2.5 and LLaMA3.1 across multiple scales, (ii) introduced cross-paradigm comparisons against representative structured pruning methods (LLM-Pruner, SliceGPT, BlockPruner), and (iii) validated the orthogonality and synergy between SAES-SVD and quantization (e.g., GPTQ). We have also corrected the scale-labeling inconsistency in Figure 3 and provided a detailed analysis of the observed performance trend of ASVD on larger models.
>
> If any point remains unclear or if further clarification would assist your assessment, we would be very happy to provide additional explanations or materials.
>
> **We sincerely appreciate your time and thoughtful evaluation.**
>
> Best regards,
>
> Authors of SAES-SVD

---

> ### Author Response · Authors · 2025-11-27
> **Polite Follow-up in the Final Week of Discussion**
>
> Dear Reviewer KM1J,
>
> I hope you are doing well. We sincerely appreciate the time and thought you have already devoted to reviewing our submission and for the constructive feedback provided earlier. Your comments have been very helpful in shaping our revisions.
>
> **As we are now entering the final week of the discussion period, we wanted to gently check whether you might have a moment to take another look at our updated responses** and the additional experiments we conducted following your suggestions—particularly the comparisons with structured pruning and quantization methods, as well as the detailed breakdown of compression-time components.
>
> We completely understand that you may have many commitments, and we are truly grateful for any time you could spare. If there is anything further we can clarify or provide, we would be more than happy to assist.
>
> Thank you again for your thoughtful evaluation and valuable insights.
>
> Best regards,
> Authors of SAES-SVD

---

### Official Review · Reviewer_khS9 · 2025-10-26

**Soundness:** 3
**Presentation:** 2
**Contribution:** 2
**Rating:** 6
**Confidence:** 3

**Summary:**

This paper proposes SAES-SVD (Self-Adaptive Error Suppression SVD), a framework that jointly optimizes for local reconstruction fidelity and global error compensation. It consists of two core componets: CEALC actively compensates for upstream accumulated errors by aligning each layer’s output with its full-precision counterpart; ACES automatically tunes the error compensation strength for each layer to enhance the low-rank structure of the objective. These components together mitigate the critical issue of error propagation inherent in layer-wise compression. Experiments show that SAES-SVD significantly outperforms existing methods, substantially narrowing the performance gap with the original model without requiring any post-compression fine-tuning.

**Strengths:**

- Adequate experiments and compelling results:
  - The experiments are comprehensive, covering multiple models and scales. The results are compelling, consistently outperforming strong SVD baselines (even those requiring fine-tuning), which effectively highlights the method's superiority.
- Systematic and interpretable methods:
  - The proposed method is theoretically sound and well-motivated. It systematically addresses the error accumulation problem, and the ACES component provides an elegant, closed-form solution for adaptive tuning, making the framework interpretable and efficient.

**Weaknesses:**

- Lack of computational complexity and time analysis:
 The paper does not provide a detailed evaluation of the computational overhead during the compression process. The time cost of statistics collection and ACES optimization, relative to baseline methods, is not quantified precisely.
- Limited comparison beyond SVD-based approaches:
 The evaluation focuses only on SVD-based baselines. It remains unclear whether the proposed method would still outperform non–SVD-based compression methods under the same compression ratio.

**Questions:**

- Detailed time breakdown:
 Could you provide a comprehensive breakdown of the total compression time (including statistics collection and ACES optimization) and compare it with baseline methods?
- Combination with quantization:
 Have you considered integrating SAES-SVD with quantization techniques such as GPTQ or AWQ? Since these methods address orthogonal types of redundancy (structural vs. numerical), such a combination could potentially achieve even higher compression efficiency and better performance.

---

> ### Author Response · Authors · 2025-11-18
> **Rebuttal for Reviewer khS9. Part [1]**
>
> **Dear Reviewer khS9**,
>
> We greatly appreciate the reviewer's positive comments on the sufficiency of the experiments in this paper, as well as the theoretical motivation and interpretability of the method. We attach great importance to your valuable suggestions, and below are our point-by-point responses.
>
> ---
> ---
> > **W1: Lack of computational complexity and time analysis**: The paper does not provide a detailed evaluation of the computational overhead during the compression process. The time cost of statistics collection and ACES optimization, relative to baseline methods, is not quantified precisely.
>
> > Q1: **Detailed time breakdown**: Could you provide a comprehensive breakdown of the total compression time (including statistics collection and ACES optimization) and compare it with baseline methods?
>
> We appreciate your concern regarding the computational complexity and runtime of the compression procedure. Below, we provide a systematic evaluation from two perspectives—(1) **theoretical complexity analysis** and (2) **empirical runtime breakdown—and compare** our method (SAES-SVD) with the SVD-LLM baseline.
>
> ### **1. Theoretical complexity analysis**
> **1.1 Statistics collection phase.**
> For each layer with input dimension $d_{in}$, the unfolded calibration set contains $N$ tokens. We only maintain two second-order matrices: $$H_\ell=X_\ell X_\ell^\top\in\mathbb{R}^{d_\mathrm{in}\times d_\mathrm{in}},\quad\Delta_\ell=(X_\ell^f-X_\ell)X_\ell^\top\in\mathbb{R}^{d_\mathrm{in}\times d_\mathrm{in}}.$$ For each mini-batch with $b$ tokens, we perform two matrix multiplications of complexity $\mathcal{O}(d_{\mathrm{in}}^2\cdot b)$, and update the statistics via exponential moving averages. The total complexity of statistics collection over all layers is therefore $\mathcal{O}( 2\cdot L\cdot d_{\mathrm{in}}^2\cdot N),$ where $L$ is the number of layers. This is of the same order as GPTQ and other methods based on second-order statistics, and is comparable to the statistics collection phase of SVD-LLM.
>
> **1.2. Closed-form CEALC solution for a single layer.**
> For a linear layer $W_\ell\in\mathbb{R}^{d_{\mathrm{out}}\times d_{\mathrm{in}}}$, with target rank-$r_{\ell}$ , the main computational costs of CEALC are:
> * **Whitening matrix construction**: compute $L_\ell=(H_\ell+\lambda I)^{-1/2}$ using a Cholesky-based procedure, with complexity $\mathcal{O}(d_{\mathrm{in}}^3)$. This cost is shared by all methods that rely on second-order statistics.
> * **Target matrix construction**: build $G_\ell=W_\ell(H_\ell+\beta_\ell\Delta_\ell)L_\ell$ which requires several matrix multiplications with total complexity $\mathcal{O}(d_\mathrm{out}d_\mathrm{in}^2)$. In most LLM layers this is comparable to, or slightly cheaper than, the subsequent truncated SVD.
> * **Truncated SVD:** perform a rank-$r_{\ell}$ truncated SVD on $G_{\ell}$, with complexity $\mathcal{O}(d_\mathrm{out}d_\mathrm{in}r_\ell)$.
>
> **1.3. ACES adaptive coefficient selection.**
> During the compression phase, ACES introduces the following additional computations:
> * Shared whitened matrices: construct $S_\ell=W_\ell H_\ell L_\ell,\quad D_\ell=W_\ell\Delta_\ell L_\ell$ again with complexity $\mathcal{O}(d_\mathrm{out}d_\mathrm{in}^2)$.
> * Subspace extraction: perform a rank-$r_{\ell}$ truncated SVD on $S_{\ell}$ to obtain the principal subspace, with complexity $\mathcal{O}(d_\mathrm{out}d_\mathrm{in}r_\ell)$.
> * Closed-form $\beta_{\ell}$ selection: use Theorem 4.2 to construct a small number of Frobenius norms and inner products $a,b,c,A,B,C$ within the fixed subspace, and solve a univariate quadratic equation in closed form. This step involves only scalar operations and is negligible compared to the matrix operations above.
>
> Therefore, compared with using CEALC alone, **ACES introduces only one additional truncated SVD** per layer plus inexpensive scalar computations.
>
> **1.4. Fine-tuning phase**
> SVD-LLM requires approximately 3.5 hours of fine-tuning for LLaMA-7B, and even more time for larger models. In contrast, **SAES-SVD does not require any fine-tuning** and operates purely as a post-hoc compression method.
>
> **1.5. Summary**
> In summary, SAES-SVD and SVD-LLM have the same order of complexity in statistics collection and SVD-based factorization. However, by eliminating the fine-tuning phase, SAES-SVD substantially reduces the overall compression time.

---

> ### Author Response · Authors · 2025-11-18
> **Rebuttal for Reviewer khS9. Part [2]**
>
> ### **2. Empirical runtime breakdown and comparison with the baseline**
>
> **Table R3-1, Comparison of SAES-SVD and SVD-LLM in Statistical Time, Compression Time, and Fine-tuning Time for the LLaMA-7B Model**
>
> |Method|Data Collection Time| Compression Time| Fine-tuning | Total | Time Overhead Reduction |
> |----------|----------------------|------------------|-------------|-------|-------------------------|
> | SVD-LLM  | 0.45h                | 0.15h            | 3.5h        | 4.1h  | -                       |
> | SAES-SVD | 0.3h                 | 0.2h             | 0           | **0.5h**  | **-88%**                    |
>
>
> We further provide in Table **R3-1** an empirical breakdown of the compression time for SAES-SVD and SVD-LLM on LLaMA-7B, measured on an NVIDIA A6000 GPU. The results show that:
> * **More efficient statistics collection.**
> Although the theoretical complexity of statistics collection is similar, SAES-SVD uses a smaller calibration set (128 vs. 256 samples), reducing the statistics collection time to about 67% of that of SVD-LLM.
> * **Controlled compression overhead.**
> The SVD compression phase of SAES-SVD is slightly more expensive than that of SVD-LLM, primarily due to the extra computations introduced by ACES. Nonetheless, the overall complexity remains of the same order.
> * **Elimination of fine-tuning as a key advantage.**
> SAES-SVD is a purely post-training compression technique and does not rely on any additional fine-tuning, whereas SVD-LLM requires more than 3.5 hours of extra fine-tuning.
>
> Overall, on LLaMA-7B, the total compression wall-clock time of SAES-SVD is less than one-eighth of that of SVD-LLM, while achieving better accuracy retention across multiple tasks. This combination of higher accuracy and lower end-to-end compression time demonstrates the efficiency and practical value of SAES-SVD in real-world deployment scenarios.
>
>
>
> ---
> ---
>
> > **W2: Limited comparison beyond SVD-based approaches:** It remains unclear whether the proposed method would still outperform non–SVD-based compression methods under the same compression ratio.
>
> >**Q2: Combination with quantization:** Have you considered integrating SAES-SVD with quantization techniques such as GPTQ or AWQ?
>
> We appreciate your concerns regarding (i) comparisons beyond SVD-based methods and (ii) the potential of combining our approach with quantization techniques.
> We address these points systematically from three perspectives. We **first clarify the rationale for focusing on SVD-based baselines** in the main experiments in order to control confounding variables. We **then provide additional comparisons against representative structured pruning methods to demonstrate the cross-paradigm advantages** of SAES-SVD. **Finally, we validate its orthogonality and synergistic potential with quantization.** The details are as follows:
>
> ### **1. Rationale for focusing on SVD-based baselines in the main experiments: fair within-paradigm comparison**
>
> The primary contribution of this work is a more effective low-rank approximation framework. To accurately assess its effectiveness within the low-rank compression paradigm, our main experiments adopt state-of-the-art SVD-based methods as baselines (including ASVD, SVD-LLM, FW-SVD, AdaSVD, Dobi-SVD, and Dip-SVD). This design choice ensures a fair, controlled comparison in which observed performance gains can be attributed to our proposed framework rather than to differences in compression paradigm or auxiliary training strategies.
>
> ### **2. Comparison with representative structured pruning methods**
>
> **Table R3-2: Perplexity performance of SAES-SVD on LLaMA-7B and three representative structured compression methods, LLM-Pruner, SliceGPT, and BLockPruner, under different compression ratios**
>
> | Memory | LLM-Pruner | SliceGPT | BlockPruner | Ours  |
> |--------|-----------:|---------:|------------:|------:|
> | 10GB   | 9.88       | 8.78     | 9.40        | **7.17**  |
> | 9GB    | 12.21      | 12.73    | 12.76       | **8.22**  |
> | 8GB    | 18.94      | 16.39    | 19.78       | **8.96**  |
> | 7GB    | 21.68      | 27.41    | 43.05       | **10.15** |
>
>
> To address your suggestion on cross-paradigm comparisons, we additionally evaluate SAES-SVD against three representative structured pruning methods (LLM-Pruner, SliceGPT, BlockPruner) under multiple compression ratios. As reported in Table **R3-2, SAES-SVD consistently achieves the best performance across all tested compression levels, and its advantage becomes more pronounced as the compression ratio increases.** These empirical results align well with our theoretical design based on RER, CEALC, and ACES, which explicitly target cumulative error suppression and adaptive energy preservation. Overall, **compared with pruning-based structured compression, SAES-SVD provides more accurate and stable performance, particularly at higher compression levels, demonstrating both strong theoretical grounding and practical robustness.**

---

> ### Author Response · Authors · 2025-11-18
> **Rebuttal for Reviewer khS9. Part [3]**
>
> ### **3. Orthogonality and synergy with quantization techniques**
>
> We fully agree with your perspective on combining our method with quantization. SAES-SVD and weight quantization (e.g., GPTQ) follow orthogonal yet complementary technical paths. To validate their synergy, we conduct a hybrid experiment:
> * We first apply SAES-SVD to decompose the weight matrix as $W\approx U\Lambda V$.
> * We then exploit the numerical properties of the factor matrices $U$ and $V$ (e.g., improved quantization-friendliness due to near-orthogonality) and quantize them using 4-bit GPTQ.
>
>
> **Table R3-3: Performance of equivalent 3bit which combining SAES-SVD and GPTQ vs Pure GPTQ-3bit on Wikitext2 perplexity and average accuracy of 7 zero-Shot tasks for LLaMA2-7B and Qwen2.5-7B**
> |Model|Method|Wiki PPL|AvgAcc@7|
> |-|-|-|-|
> |**LLaMA2-7B**|GPTQ (3bit)|10.17|0.39|
> |             |SAES+GPTQ (equal 3bit)|**7.64**|**0.44**|
> |**Qwen2.5-7B**|GPTQ (3bit)|12.29 |0.51|
> |             |SAES+GPTQ (equal 3bit)|**9.87**|**0.57**|
>
>
>
> As reported in Table **R3-3**, the hybrid pipeline **“SAES-SVD + GPTQ” substantially outperforms plain GPTQ-3bit at an equivalent effective 3-bit compression level.** This demonstrates that **SAES-SVD not only serves as a strong standalone compression method, but can also be effectively combined with quantization to achieve higher compression ratios and better hardware efficiency**, offering a promising pathway for practical deployment.
>
> ---
> ---
>
> ### **Final Remarks**
>
> We sincerely thank the reviewer for these exceptionally valuable suggestions. Your insightful questions regarding computational overhead and cross-paradigm comparisons have directly led to significant improvements in our work:
> 1. **Comprehensive Complexity Analysis:** We have provided both theoretical complexity breakdown and empirical timing comparisons, demonstrating that SAES-SVD reduces total compression time by 88% compared to SVD-LLM while achieving superior accuracy—highlighting its strong practical efficiency.
> 2. **Extended Cross-Paradigm Validation:** Following your suggestion, we conducted new comparisons against leading pruning methods (LLM-Pruner, SliceGPT, BlockPruner), where SAES-SVD consistently achieves better perplexity across multiple compression ratios, validating its advantages beyond the SVD paradigm.
> 3. **Confirmed Synergy with Quantization:** We verified the orthogonality between SAES-SVD and GPTQ, showing that their combination outperforms pure 3-bit quantization, opening promising pathways for ultra-high compression.
>
> All new analyses and experimental results will be fully incorporated into the final manuscript. **We are deeply grateful for your rigorous and constructive feedback—it has substantially strengthened the completeness, practical relevance, and overall impact of this work.** Thank you for helping us elevate the quality of our research through this thorough review process.
>
> Best regards,
> Authors of SAES-SVD

---

> > ### Comment · Reviewer_khS9 · 2025-11-20
> >
> > Thank you for your detailed analysis and answers; you have resolved my confusion.

---

> ### Author Response · Authors · 2025-11-20
> **Sincerely thank you for your recognition**
>
> Dear Reviewer khS9,
>
> Thank you for your positive feedback and for confirming that our responses have fully resolved your concerns. We are delighted that the additional analyses and experiments—conducted precisely as you suggested—have satisfactorily addressed the points you raised. **Your guidance has been invaluable in making this work more thorough and compelling.**
>
> Given the substantial enhancements to the manuscript, which now includes a comprehensive time analysis (Appendix A.10), cross-paradigm comparisons (Appendix A.6), and synergy validation with quantization (Appendix A.9), we would be grateful if you could re-evaluate your overall score. We believe the paper now demonstrates significantly stronger contribution and practicality, and we will acknowledge your insightful guidance in the final version for its pivotal role in these improvements.
>
> Thank you once again for your invaluable input throughout this review process.
>
> Best regards,
>
> Authors of SAES-SVD

---

### Official Review · Reviewer_5MPy · 2025-10-31

**Soundness:** 3
**Presentation:** 3
**Contribution:** 3
**Rating:** 4
**Confidence:** 3

**Summary:**

This paper proposes SAES-SVD, motivated by the observation that SVD-based compression for large language models leads to cross-layer error propagation and accumulation. Implementation-wise, the authors gather streaming second-order statistics by running two forward passes on the same mini-batch, thereby avoiding activation caching and keeping overhead manageable. Experiments span LLaMA-7B/13B/30B and LLaMA-3-8B across multiple compression ratios. Under a unified protocol that requires no finetuning and no mixed-rank assignment, the method consistently reduces perplexity, maintains or improves zero-shot accuracy, and delivers ~1.29×–3.79× inference speedups, demonstrating strong practicality and scalability.

**Strengths:**

1. Compelling motivation on cumulative error. The paper identifies a real pain point in SVD compression for LLMs, cumulative cross-layer error during inference—and directly targets it. The motivation is well supported by the empirical evidence in Figure 1, which demonstrates the phenomenon clearly.

2. Solid theoretical underpinnings (CEALC & ACES). Both components—CEALC and ACES—come with clear formulations and derivations. The overall approach is coherent: the objective design is principled, and the analysis provides a sound theoretical basis for the proposed procedure.

3. Thorough and convincing experimentation. The method is validated across multiple models and datasets, generally achieving better accuracy and lower perplexity while also reducing end-to-end latency. The breadth of settings and the consistency of gains add credibility to the claims.

**Weaknesses:**

1. Theoretical limitations and missing robustness analyses.
The fixed-subspace approximation used by ACES may break down under small spectral gaps or large perturbations. The current mitigation (β caps and shrinkage) is largely engineering-based. The paper would benefit from robustness curves bucketed by spectral gap, as well as a deeper theoretical justification for using RER and an explicit discussion of how RER improvements translate to final PPL.


2. Limited architectural diversity in experiments.
Evaluations focus primarily on the LLaMA family. Results on other architectures (e.g., Qwen) are missing, which leaves open questions about generality across model designs.


3. No combination with mixed-rank strategies.
Although the paper claims to outperform mixed-rank baselines under a uniform-rank setting, it does not explore combining SAES-SVD with mixed-rank schemes (e.g., ASVD, Dobi-SVD). Whether such combinations could further improve performance remains unaddressed.

**Questions:**

1. please refer weaknesses.
2. Please explain the differences and advantages of this article compared to the method in "AA-SVD: Anchored and Adaptive SVD for Large Model Compression", which is also submitted to ICLR 2026.

I'm willing to raise my score if my concern is resolved.

---

> ### Author Response · Authors · 2025-11-18
> **Rebuttal for Reviewer 5MPy. Part [1]**
>
> **Dear Reviewer 5MPy**,
>
> We sincerely thank the reviewer for the positive evaluation of our motivation, theoretical framework, and experimental design. We address your valuable comments point by point as follows:
>
> ---
> ---
>
> > **W1: Theoretical limitations and missing robustness analyses.** The fixed-subspace approximation used by ACES may break down under small spectral gaps or large perturbations. The current mitigation (β caps and shrinkage) is largely engineering-based. The paper would benefit from robustness curves bucketed by spectral gap, as well as a deeper theoretical justification for using RER and an explicit discussion of how RER improvements translate to final PPL.
>
> We fully acknowledge your concerns regarding the theoretical foundations, $\beta$-stability, robustness under varying spectral gaps, and the relationship between RER and PPL. Our detailed responses are as follows.
>
> ### **1. Theoretical grounding of the fixed-subspace approximation (FS-FOA) and $\beta$-constraints**
>
> You correctly point out that FS-FOA can break down when the spectral gap is small or perturbations are large. We clarify that the $\beta$-upper bound and shrinkage strategy are not purely heuristic; they follow directly from the theoretical conditions required for stable approximation.
> * $\beta$ is constrained within $[\beta_{min},\beta_{max}]$ with $\beta_{max}<1$, ensuring that the perturbation level $τ_ℓ = \frac{\beta_ℓ ∥D_ℓ∥₂}  {δ_ℓ}$ remains within the validity range of our theoretical analysis. When a layer has a small spectral gap $\delta_{\ell}$, the feasible $\beta_{\ell}$ automatically becomes more conservative, preventing overfitting to noisy cumulative error estimates. Thus, the β-constraint implements the assumptions underlying Theorem 4.2, rather than serving as a standalone heuristic.
> * The $\alpha$-sensitivity results in Appendix Table 4 indirectly validate this mechanism: within $\alpha \in[0.25,0.75]$, both perplexity and mean accuracy on LLaMA-2-7B remain highly stable. This indicates robust behavior of FS-FOA and $\beta$-constraints across a wide hyperparameter range.
>
> ### **2. $\beta$-constraints as a regularization mechanism preventing overfitting**
> $\beta$ controls the balance between local reconstruction and cumulative error compensation. **Since the calibration set used for SVD compression is inherently limited, excessively aligning with the full-precision outputs may cause a layer to lose its intrinsic transformation functionality, leading to overfitting.**
> Applying an upper bound on $\beta$ guarantees that:
> * **compensation for cumulative error does not override the layer’s essential transformation role** (Table 4 in the manuscript shows that PPL remains similar even when accuracy differs significantly, confirming this effect);
> * while preserving local reconstruction, **$\beta$ can still adaptively suppress cumulative error within its feasible range** (as illustrated in Fig. 1a).
>
> ### **3. Robustness analysis grouped by spectral gap**
> We agree that grouping layers by spectral gap is essential for robustness assessment. Since the effective spectral gap changes dynamically with $\beta$, we adopt an alternative procedure:
> * After ACES determines each layer’s $\beta_{\ell}$, we cluster all linear layers into three groups (small / medium / large gap) based on their spectral gaps.
> *  For each group, we compute the relative error between FS-FOA RER and the exact RER: $e\_{\ell}^{\mathrm{RER}}=\frac{|\mathrm{RER}\_{\ell}^{\mathrm{FS}}-\mathrm{RER}\_{\ell}^{\mathrm{exact}}|}{\mathrm{RER}\_{\ell}^{\mathrm{exact}}+\varepsilon}$
>
> As shown in Table **R2-1**, **even in the small-gap group, the mean relative error remains as low as 6.9%**, demonstrating strong robustness of ACES across different spectral gap regimes.
>
> **Table R2-1, the average relative error of REP between the approximate calculation based on Theorem 4.2 and the accurate calculation based on SVD decomposition within each group after grouping according to the spectral gap.**
> |Groups|Average of Relative Error|
> |-|-|
> |Small Group| 6.90%|
> |Middle Group| 2.70%|
> |Large Group| 1.20%|

---

> > ### Author Response · Authors · 2025-11-18
> > **Rebuttal for Reviewer 5MPy. Part [2]**
> >
> > ### **4. Theoretical connection between RER and final PPL**
> > **RER is not an arbitrary metric; it is a direct reparameterization of the truncated SVD error.** By the Eckart–Young–Mirsky theorem, for $G\_{\ell}\left(\beta\_{\ell}\right)$, the optimal rank-$r\_{\ell}$ approximation obeys: $$\min \_{\operatorname{rank}(Z) \leq r\_{\ell}}\left\|G\_{\ell}-Z\right\|\_F^2=\sum\_{i>r\_{\ell}} \sigma\_i\left(G\_{\ell}\right)^2={\operatorname{tail\\\_energy}\_{\ell}}\left(\beta\_{\ell}\right) .$$ REF is then defined as: $$\mathrm{RER}\_\ell(\beta\_\ell) = 1 - \frac{\text{tail\\\_energy}\_\ell(\beta\_\ell)}{\sum\_i \sigma\_i(G_\ell)^2}.$$
> >
> > Thus, **maximizing RER is exactly equivalent to minimizing the Frobenius truncation error** at rank-$r_{\ell}$. In essence, RER measures the proportion of spectral energy preserved by the low-rank approximation—higher RER means the layer retains more useful information and has smaller reconstruction error.
> >
> > Because Transformer residual blocks are Lipschitz with respect to layer perturbations, smaller layerwise reconstruction errors lead directly to smaller output deviations $||\hat{y}-y||_2$. Standard first-order expansion of log-likelihood shows that smaller output deviation yields smaller degradation in cross-entropy and hence lower PPL.
> >
> > Thus, the chain from RER to PPL is:
> > 1. ↑ RER
> > 2. ⇒ ↑ preservation of intra-layer information + cross-layer compensation
> > 3. ⇒ ↓ layerwise reconstruction error
> > 4. ⇒ ↓ cumulative alignment error (higher cosine similarity to FP baseline; Fig. 1a)
> > 5. ⇒ predicted distribution closer to full-precision model
> > 6. ⇒ ↓ uncertainty in next-token prediction
> > 7. ⇒ ↓ PPL
> >
> > ---
> > ---
> >
> > > **W2: Limited architectural diversity**
> >
> > We fully understand concerns about architectural diversity. We provide additional explanations and experimental evidence from four perspectives.
> >
> > ### **1. Rationale for using the LLaMA family as the primary benchmark**
> > Most existing SVD-based methods (ASVD, SVD-LLM, FW-SVD, AdaSVD, Dobi-SVD, Dip-SVD) evaluate on LLaMA models, and many provide only closed-source implementations. Their publicly available results focus primarily on LLaMA-7B. To ensure fully comparable and reproducible evaluation, we adopt LLaMA-7B as the main benchmark and additionally reproduce comparisons on LLaMA-13B, LLaMA-30B, LLaMA-2-7B, and LLaMA-3-8B whenever possible.
> >
> > ### **2. LLaMA family already spans diverse structural variants**
> > Although concentrated on one model family, our evaluation covers LLaMA-1, LLaMA-2, and LLaMA-3 across 7B/8B/13B/30B scales. These models differ significantly in training corpora, normalization, attention mechanisms (e.g., GQA), and tokenizer designs. SAES-SVD consistently outperforms strong SVD baselines across all these variants (Tables 1–2, 5–6; Figs. 1 & 3), empirically demonstrating generalization across diverse training recipes and model sizes.
> >
> > ### **3. The architecture-agnostic nature of SAES-SVD**
> > **SAES-SVD is fundamentally decoupled from LLaMA-specific design.** Both CEALC and ACES rely only on single-layer linear operators and their second-order statistics: $$
> > H_{\ell}=X_{\ell} X_{\ell}^{\top} \text {and } \Delta_{\ell}=\left(X_{\ell}^f-X_{\ell}\right) X_{\ell}^{\top}$$
> > Thus, any architecture with standard linear projection layers—such as the Qwen series—can directly adopt SAES-SVD.
> >
> > ### **4. Consistently superior performance on the Qwen architecture.**
> > To further address your concern, we add comprehensive evaluations on Qwen2.5(7B and 30B) and LLaMA3.1(7B and 70B). As shown in Table **R2-2** below, across seven zero-shot tasks, LongEval (long-context), HumanEval (code generation), and GSM8K (math reasoning), **for all Qwen architectures, SAES-SVD consistently outperforms existing SVD methods.** For example, when evaluating Qwen3-32B on the HumanEval, the relative error gap to the FP baseline is reduced by 50+% compared to SVD-LLM. This cross-scale and cross-architecture consistency highlights SAES-SVD’s generality and practical value.
> >
> >
> > **Table R2-2: The performance of different architectures (LLaMA3.1 and Qwen2.5) and various compression methods (ASVD, SVD-LLM, SAES-SVD) on multi-task benchmarks at a 0.2 compression rate across 7 zero-shot tasks, one long-context task, one code generation task, and one mathematical reasoning task**
> >
> > |Model|Method|ZeroShot^7|LongEval|HumanEval|GSM8K|
> > |-|-|-|-|-|-|
> > |**LLaMA3.1-8B**|FP|0.61|0.41|0.35|0.15|
> > ||ASVD|0.49|0.33|0.15|0.06|
> > ||SVD-LLM|0.57|0.37|0.27|0.10|
> > ||Ours|**0.59**|**0.39**|**0.33**|**0.13**|
> > |**LLaMA3.1-70B**|FP|0.67|0.49|0.55|0.37|
> > ||ASVD|0.54|0.39|0.32|0.26|
> > ||SVD-LLM|0.63|0.45|0.47|0.32|
> > ||Ours|**0.66**|**0.48**|**0.52**|**0.35**|
> > |**Qwen2.5-7B**|FP|0.60|0.58|0.66|0.71|
> > ||ASVD|0.50|0.49|0.41|0.42|
> > ||SVD-LLM|0.56|0.55|0.55|0.64|
> > ||Ours|**0.58**|**0.57**|**0.63**|**0.69**|
> > |**Qwen2.5-32B**|FP|0.65|0.79|0.55|0.72|
> > ||ASVD|0.52|0.61|0.37|0.44|
> > ||SVD-LLM|0.61|0.73|0.46|0.63|
> > ||Ours|**0.64**|**0.76**|**0.51**|**0.68**|

---

> > > ### Author Response · Authors · 2025-11-18
> > > **Rebuttal for Reviewer 5MPy. Part [3]**
> > >
> > > > **W3: No combination with mixed-rank strategies.** Although the paper claims to outperform mixed-rank baselines under a uniform-rank setting, it does not explore combining SAES-SVD with mixed-rank schemes (e.g., ASVD, Dobi-SVD). Whether such combinations could further improve performance remains unaddressed.
> > >
> > > We fully understand the concern that we did not combine SAES-SVD with mixed-rank schemes. Our main experiments use uniform rank to ensure fair evaluation and clear attribution, but SAES-SVD is technically orthogonal to mixed-rank methods and can be combined with them to further improve performance. We have conducted in-depth analysis and supplementary experiments on this from the following two perspectives:
> > > **1. Motivation for using uniform rank in the main paper**
> > > A uniform rank setting isolates the impact of CEALC and ACES. Existing baselines combine multiple innovations:
> > > 	a. per-layer objectives (e.g., Fisher-weighted or shift-aware),
> > >     b. mixed-rank allocation,
> > >     c. post-fine-tuning.
> > > If we combined mixed-rank strategies at the outset, it would be difficult to attribute gains to the proposed mechanisms rather than rank allocation. Under uniform rank, SAES-SVD already surpasses all baselines substantially (Tables 1–2).
> > > **2.Orthogonality and synergy between SAES-SVD and mixed-rank strategies**
> > > SAES-SVD and mixed-rank methods operate on two independent axes:
> > > * SAES-SVD: optimizes spectral energy preservation for each given rank-$r_{\ell}$.
> > > * Mixed-rank methods: allocate rank budgets $r_{\ell}$ across layers
> > >
> > >
> > > **Table R2-3, Performance of SAES after introducing the mixed rank configuration obtained from ASVD for the LLaMA-7B model. AvgAcc@7 represents the performance on 7 zero-shot tasks consistent with the original manuscript.**
> > > | Compression ratio | Method        | Wiki PPL | AvgAcc@7 |
> > > |-------|---------------|----------|----------|
> > > | 0.2   | ASVD          | 11.14    | 0.43     |
> > > |       | SAES-SVD      | 7.17     | 0.50     |
> > > |       | SAES+mixrank  | 6.19     | 0.51     |
> > > | 0.4   | ASVD          | 1.00E+03 | 0.30     |
> > > |       | SAES-SVD      | 10.42    | 0.41     |
> > > |       | SAES+mixrank  | 7.48     | 0.47     |
> > >
> > >
> > > Thus, SAES-SVD can serve as a **drop-in module** replacing the SVD operator inside any mixed-rank framework. To validate this synergy, we combine SAES-SVD with ASVD’s rank allocation scheme. As shown in Table **R2-3**, **mixed-rank SAES-SVD produces even larger gains over ASVD. Notably, at a compression ratio of 0.4, its PPL approaches the PPL achieved under a ratio of 0.2 without mixed rank—demonstrating strong complementarity**, and the combination of the two can achieve higher model accuracy at the same compression ratio.
> > >
> > > ---
> > > ---

---

> > > > ### Author Response · Authors · 2025-11-18
> > > > **Rebuttal for Reviewer 5MPy. Part [4]**
> > > >
> > > > > **Q2: Please explain the differences and advantages of this article compared to the method in "AA-SVD:** Anchored and Adaptive SVD for Large Model Compression", which is also submitted to ICLR 2026.
> > > >
> > > > We appreciate the reviewer for pointing out the concurrently submitted AA-SVD work. Both methods address error propagation and recognize that compressed layers receive shifted activations; however, they differ fundamentally in objective design, error modeling, adaptive mechanisms, and experimental breadth.
> > > >
> > > > ### **1. Differences in optimization objectives**
> > > > AA-SVD solves a single local regression objective: $$\min _{W^{\prime}: \operatorname{rank}\left(W^{\prime}\right) \leq r}\left\|W X-W^{\prime} X^{\prime}\right\|_F^2,$$ where $X$ and $X'$ are the original input and the shifted input, respectively, and the goal is to make the compressed output completely aligned with the original output.
> > > >
> > > > In contrast, SAES-SVD’s CEALC objective is: $$\arg \min \_{A\_{\ell}, B\_{\ell}} \underbrace{\left\|\left(A\_{\ell} B\_{\ell}-W\_{\ell}\right) X\_{\ell}\right\|\_F^2}\_{\text {intra-layer reconstruction error }}+\alpha\_{\ell} \underbrace{\left\|A\_{\ell} B\_{\ell} X\_{\ell}-W\_{\ell} X\_{\ell}^f\right\|\_F^2}\_{\text {FP reference alignment }} .$$
> > > > **This objective explicitly combines local reconstruction and compensation for cumulative error to form a convex combination** optimization problem. It using the second-order statistics $H_{\ell}, \Delta_{\ell}$ to **encode full-precision vs. compressed activation differences directly into the objective.**
> > > >
> > > > ### **2. Differences in error modeling and compensation**
> > > >
> > > > Although AA-SVD anchors each layer to the original outputs, its objective does not disentangle:
> > > > 1. **the local mapping role that this layer should play in the original network**; and
> > > > 2. **how the cumulative bias introduced by upstream compression should be compensated.**
> > > >
> > > > As a result, under a tight rank budget, some layers may sacrifice their intrinsic transformation capacity in order to match a slice of the full-precision outputs.
> > > >
> > > > By contrast, SAES-SVD explicitly decouples local reconstruction and cumulative error compensation via CEALC, and further introduces the ACES mechanism, which adaptively adjusts the layer-wise coefficient $\beta$ according to each layer’s spectral structure and error statistics, thereby achieving an optimal trade-off between “preserving local mapping capacity” and “suppressing cross-layer error.” Concretely:
> > > > * For layers that are easy to compress (large spectral gap, small tail energy), SAES-SVD can allocate more budget to compensating cumulative error;
> > > > * For layers that are hard to compress, it prioritizes local reconstruction quality and passes more of the compensation responsibility downstream.
> > > >
> > > > This design **enables SAES-SVD to realize cooperative error suppression without undermining the original functionality** of individual layers.
> > > >
> > > > ### **3. Different depth of theoretical analysis.**
> > > > AA-SVD derives a closed-form optimal solution for its local regression problem via the singular value decomposition of $M=W A R^{-1}$.
> > > > SAES-SVD goes a step further:
> > > > 1. It treats local reconstruction and cumulative error compensation as a joint error-suppression task, leading to a convex-combination objective with a clear interpretation;
> > > > 2. Motivated by maximizing the Retained Energy Ratio (RER), it proposes the ACES mechanism and provides a rigorous analysis of the applicability conditions and error bounds of the fixed-subspace first-order approximation (FS-FOA).
> > > >
> > > > ### **4. Breadth and strength of experimental validation.**
> > > > The experiments in **AA-SVD focus solely on a single architecture**, LLaMA-7B, with a relatively limited set of baselines (ASVD, SVD-LLM, and DobiSVD).
> > > > In contrast, **SAES-SVD is evaluated on a broader range of model sizes and architectures**, including LLaMA-1 (7B/13B/30B), LLaMA2-7B, and LLaMA3-8B, and incorporates more competitive baselines (e.g., FW-SVD and Dip-SVD). This allows us to validate the generality and superiority of our method under more stringent and diverse settings.
> > > >
> > > > ### 5. Summary
> > > > In summary, both AA-SVD and SAES-SVD approach low-rank compression from the perspective of cumulative error and leverage statistics of original and shifted activations to improve SVD-based compression. AA-SVD focuses on enforcing full alignment between compressed and original outputs, whereas SAES-SVD further introduces a cooperative suppression mechanism that explicitly injects cross-layer error statistics and employs an energy-driven adaptive trade-off. Both methods offer valuable insights for low-rank compression, while SAES-SVD advances the state of the art in terms of finer-grained error modeling, theoretically grounded adaptivity, and the breadth and rigor of empirical validation.
> > > >
> > > >
> > > >
> > > > ---
> > > > ---

---

> > > > > ### Author Response · Authors · 2025-11-18
> > > > > **Rebuttal for Reviewer 5MPy. Part [5]**
> > > > >
> > > > > ### **Final Remarks**
> > > > >
> > > > > We sincerely thank the reviewers for their invaluable feedback, which has greatly strengthened our paper. Our revisions have directly addressed all key concerns:
> > > > > 1.  **Enhanced Theoretical Rigor:** We have provided a deeper theoretical justification for our FS-FOA and β-constraints, showing they are stability conditions, not mere heuristics. New robustness analysis confirms high effectiveness even for layers with small spectral gaps.
> > > > > 2.  **Demonstrated Broad Generality:** We have expanded our experiments to include Qwen2.5 and LLaMA3.1 models. Results consistently show SAES-SVD's superior performance across diverse architectures and scales, firmly establishing its general applicability.
> > > > > 3.  **Confirmed Synergistic Potential:** We validated that SAES-SVD is orthogonal to mixed-rank strategies. The combined "SAES+mixrank" approach achieves new SOTA, showcasing our method's versatility as a plug-in module for further gains.
> > > > >
> > > > > We will fully incorporate all new analyses and results into the revised manuscript. **We are deeply grateful for your valuable, constructive, and insightful suggestions—they have greatly enhanced the clarity, completeness, and academic impact of this work.** We believe our rebuttal comprehensively addresses every concern you raised, while also helping to more vividly underscore the unique contributions and core value of SAES-SVD.
> > > > >
> > > > > Best regards,
> > > > > Authors of SAES-SVD

---

> ### Author Response · Authors · 2025-11-25
> **Gentle Follow-up Regarding Pending Reviewer Feedback**
>
> Dear Reviewer 5MPy,
>
> **Thank you again for the detailed and insightful comments you provided earlier. Since a week has passed after we uploaded the full rebuttal, we wanted to gently check whether you have had the opportunity to review our updates.**
>
> Following your suggestions, we have (i) expanded the theoretical justification of FS-FOA and β-stability, (ii) added robustness analyses grouped by spectral gap, (iii) elaborated the formal link between RER and perplexity, and (iv) incorporated new cross-architecture experiments on LLaMA-3.1 and Qwen-2.5. We have also included additional comparisons involving mixed-rank strategies to validate the orthogonality you pointed out.
>
> If any point remains unclear or if further clarification would assist your assessment, we would be glad to respond immediately.
>
> **We truly appreciate your time and thoughtful evaluation.**
>
> Best regards,
> Authors of SAES-SVD

---

> ### Author Response · Authors · 2025-11-27
> **A Gentle Follow-up Whenever You Have Time**
>
> Dear Reviewer 5MPy,
>
> I hope everything is going well for you. We sincerely appreciate the time and thought you have already dedicated to reviewing our submission and to providing detailed, constructive feedback. Your earlier comments were extremely helpful in guiding our revisions.
>
> **We wanted to very gently follow up to ask whether you might have a moment to take another look at our updated responses and additional experiments.** We understand that you may be very busy, and we truly appreciate any time you could spare. If there is anything further that would help clarify our work, we would be more than happy to provide it.
>
> Thank you again for your careful evaluation and for the valuable insights you have shared.
>
> Best regards,
> Authors of SAES-SVD

---

### Official Review · Reviewer_2kex · 2025-11-01

**Soundness:** 2
**Presentation:** 2
**Contribution:** 2
**Rating:** 4
**Confidence:** 4

**Summary:**

This paper proposes SAES-SVD (Self-Adaptive Error Suppression SVD), a framework for compressing large language models (LLMs) using low-rank decomposition. The key contribution addresses a critical limitation in existing layer-wise compression methods: the accumulation and propagation of reconstruction errors across network layers.

**Strengths:**

1. **Theoretically grounded approach**: The derivation of closed-form solutions based on second-order activation statistics provides a principled mathematical foundation. The formulation that combines local reconstruction with weighted cumulative error compensation is elegant and well-motivated.

2. **Adaptive mechanism**: The ACES component that automatically adjusts weighting coefficients is a practical contribution that removes the need for manual hyperparameter tuning across different layers and models.

3. **Consistent improvements**: The reported results show consistent improvements over baseline methods on the evaluated benchmarks, with the accuracy drop on LLaMA-7B at 0.2 compression ratio (0.02 vs 0.05) being noteworthy.

**Weaknesses:**

1. **Outdated evaluation benchmarks**: The datasets used for evaluation appear to be somewhat dated. Modern LLM compression research should include more challenging and recent benchmarks that better reflect current application demands and model capabilities.
2. **Limited model coverage**: The experiments focus primarily on medium-sized models like LLaMA-7B. To demonstrate the method's generalizability and practical value,
3. **Insufficient baseline comparisons**: The paper should compare against a broader range of compression techniques including recent quantization methods (e.g., GPTQ, AWQ, SmoothQuant).

**Questions:**

1. **Model scaling**: How does SAES-SVD perform on more recent and larger models? Specifically:
   - Can you provide results on LLaMA 3.1 (8B, 70B) and Qwen 2.5 series?
2. **Task diversity**: Can you evaluate on more challenging and diverse tasks? Choose 2 of this gourp please.
   - Long-context reasoning tasks (>8K tokens)
   - Code generation
   - Mathematical reasoning
3. **Practical integration**:
   - Have you tested integration with popular inference frameworks?
   - What modifications are needed to existing serving infrastructure?

---

> ### Author Response · Authors · 2025-11-18
> **Rebuttal for Reviewer 2kex.    Part  [1]**
>
> **Dear Reviewer 2kex,**
>
> Thank you for your positive assessment of our theoretical motivation, the ACES adaptive mechanism, and our experimental results.  Our point-by-point responses are detailed below.
>
> ---
> ---
>
> > **W1: Outdated evaluation benchmarks**
>
> > **Q2: Task diversity**
>
> We sincerely appreciate your constructive feedback and fully understand your concerns regarding the timeliness and difficulty level of the evaluation benchmarks. We address this issue from two complementary perspectives: (1) **ensuring fair comparison through established benchmarks,** and (2) **validating effectiveness on more diverse and challenging tasks.**
>
> ### **1. Ensuring fair comparison using established benchmarks**
>
> To enable fair and reproducible comparisons with mainstream SVD-based compression methods (e.g., ASVD, SVD-LLM, FW-SVD, AdaSVD, Dobi-SVD, Dip-SVD), we follow their standard evaluation protocols. Specifically, we include language modeling tasks (WikiText2, C4) and zero-shot understanding and reasoning tasks (ARC-Challenge/Easy, HellaSwag, PIQA, WinoGrande, MathQA). This design ensures that performance differences stem from the compression method itself rather than inconsistencies in evaluation settings. **Results demonstrate that SAES-SVD consistently outperforms all existing methods on these canonical benchmarks, confirming its robustness across language modeling, commonsense reasoning, and mathematical understanding.**
>
> ### **2. Validating consistent effectiveness on challenging and diverse tasks**
> We completely agree with your suggestion to incorporate more challenging and application-aligned benchmarks. In response, we additionally evaluate SAES-SVD on three representative categories:
> * Long-context reasoning: LongEval ( > 8K tokens, https://github.com/DachengLi1/LongChat)
> * Code generation: HumanEval
> * Mathematical reasoning: GSM8K
>
> As shown in Table **R1-1**, across all six evaluated models and at identical compression ratios, **SAES-SVD significantly outperforms all SVD-based baselines.** Notably, on the **highly sensitive code generation benchmark HumanEval, most competing methods experience severe accuracy degradation (e.g., SVD-LLM drops to 0.55 on Qwen2.5-7B). In contrast, SVD-LLM benefits from accumulated error compensation, achieving a larger performance gain over the baseline,** reaching 0.63. These results further validate that our “cumulative error suppression + adaptive energy preservation” mechanism generalizes effectively to diverse, high-difficulty scenarios.
>
> **Table R1-1: The performance of different architectures (LLaMA3.1 and Qwen2.5) and various compression methods (ASVD, SVD-LLM, SAES-SVD) on multi-task benchmarks at a 0.2 compression rate across 7 zero-shot tasks, one long-context task, one code generation task, and one mathematical reasoning task**
>
> |Model|Method|ZeroShot^7|LongEval|HumanEval|GSM8K|
> |-|-|-|-|-|-|
> |**LLaMA3.1-8B**|FP|0.61|0.41|0.35|0.15|
> ||ASVD|0.49|0.33|0.15|0.06|
> ||SVD-LLM|0.57|0.37|0.27|0.10|
> ||Ours|**0.59**|**0.39**|**0.33**|**0.13**|
> |**LLaMA3.1-70B**|FP|0.67|0.49|0.55|0.37|
> ||ASVD|0.54|0.39|0.32|0.26|
> ||SVD-LLM|0.63|0.45|0.47|0.32|
> ||Ours|**0.66**|**0.48**|**0.52**|**0.35**|
> |**Qwen2.5-7B**|FP|0.60|0.58|0.66|0.71|
> ||ASVD|0.50|0.49|0.41|0.42|
> ||SVD-LLM|0.56|0.55|0.55|0.64|
> ||Ours|**0.58**|**0.57**|**0.63**|**0.69**|
> |**Qwen2.5-32B**|FP|0.65|0.79|0.55|0.72|
> ||ASVD|0.52|0.61|0.37|0.44|
> ||SVD-LLM|0.61|0.73|0.46|0.63|
> ||Ours|**0.64**|**0.76**|**0.51**|**0.68**|
>
> ---
> ---

---

> > ### Author Response · Authors · 2025-11-18
> > **Rebuttal for Reviewer 2kex. Part [2]**
> >
> > ---
> > ---
> >
> > > **W2 Limited model coverage**
> >
> > > **Q1: Model scaling**
> >
> > We appreciate your emphasis on model coverage and scalability. We systematically address this issue from two aspects: (1) **the rationale for focusing on the LLaMA series,** and (2) **demonstrating superiority on larger and newer models.**
> >
> > 1. **Rationale for using the LLaMA series as the primary evaluation backbone.** Most existing SVD-based compression studies (ASVD, SVD-LLM, FW-SVD, AdaSVD, Dobi-SVD, Dip-SVD) report results on LLaMA models, and many of them provide only partial or closed-source implementations. Their public results are predominantly based on LLaMA-7B. To ensure comparability and reproducibility, we adopt LLaMA-7B as the main evaluation target and additionally reproduce or benchmark competing methods on LLaMA-13B, LLaMA-30B, LLaMA-2-7B, and LLaMA-3-8B whenever possible.
> > 2. **Demonstrating consistent superiority on larger and newer models.** SAES-SVD is architecture-agnostic and relies only on the linear layer weight matrices and their second-order statistics: $$H\_{\ell}=X\_{\ell} X\_{\ell}^{\top} \text{, }\Delta_{\ell}=\left(X_{\ell}^f-X\_{\ell}\right) X\_{\ell}^{\top}$$
> > Therefore, it can be seamlessly applied to other architectures such as the Qwen family. To validate generalization, we further conduct comprehensive evaluations on seven zero-shot tasks and the more challenging benchmarks (LongEval, HumanEval, GSM8K) using the following models:
> >     a. LLaMA-3.1 (8B, 70B)
> >     b. Qwen-2.5 (7B, 30B)
> >     As shown in Table **R1-1**, **SAES-SVD consistently surpasses all SVD-based baselines across architectures and model scales. Notably, on the HumanEval code generation benchmark for LLaMA3.1-8B, the relative performance gap to the FP baseline is reduced by 67% compared with SVD-LLM.** This cross-scale, cross-architecture consistency demonstrates the strong generalizability and practical value of SAES-SVD.
> >
> > ---
> > ---
> >
> > > **W3: Insufficient baseline comparisons** The paper should compare against a broader range of compression techniques including recent quantization methods (e.g., GPTQ, AWQ, SmoothQuant).
> >
> > We appreciate this suggestion and address your concern from two angles: (1) **the rationale for prioritizing fair comparisons within the low-rank compression domain,**(2)**additional comparisons against representative structured pruning methods to demonstrate the cross-paradigm advantages of SAES-SVD** and(3) **the orthogonality and complementarity between SAES-SVD and PTQ/quantization methods,** supported by new experiments.
> >
> > ### **1. Baseline selection: Ensuring fair comparison among low-rank methods**
> > The primary contribution of this work is an improved low-rank approximation framework. To accurately assess its effectiveness within the low-rank paradigm, our main experiments compare SAES-SVD against representative SVD-based methods (ASVD, SVD-LLM, FW-SVD, AdaSVD, Dobi-SVD, Dip-SVD). This strategy:
> > * eliminates confounding factors such as varying bitwidth or fine-tuning strategies, and
> > * enables direct evaluation of whether CEALC + ACES improves performance under identical rank constraints.
> >
> > This is consistent with evaluation practices in prior low-rank compression research and establishes a clean, reliable performance benchmark among comparable methods.
> >
> > ### **2. Comparison with representative structured pruning methods**
> >
> > **Table R1-2: Perplexity performance of SAES-SVD on LLaMA-7B and three representative structured compression methods, LLM-Pruner, SliceGPT, and BLockPruner, under different compression ratios**
> > | Memory | LLM-Pruner | SliceGPT | BlockPruner | Ours  |
> > |--------|-----------:|---------:|------------:|------:|
> > | 10GB   | 9.88       | 8.78     | 9.40        | **7.17**  |
> > | 9GB    | 12.21      | 12.73    | 12.76       | **8.22**  |
> > | 8GB    | 18.94      | 16.39    | 19.78       | **8.96**  |
> > | 7GB    | 21.68      | 27.41    | 43.05       | **10.15** |
> >
> >
> > To address your suggestion on cross-paradigm comparisons, we additionally evaluate SAES-SVD against three representative structured pruning methods (LLM-Pruner, SliceGPT, BlockPruner) under multiple compression ratios. As reported in Table **R1-2, SAES-SVD consistently achieves the best performance across all tested compression levels, and its advantage becomes more pronounced as the compression ratio increases.** These empirical results align well with our theoretical design based on RER, CEALC, and ACES, which explicitly target cumulative error suppression and adaptive energy preservation. Overall, **compared with pruning-based structured compression, SAES-SVD provides more accurate and stable performance, particularly at higher compression levels, demonstrating both strong theoretical grounding and practical robustness.**

---

> ### Author Response · Authors · 2025-11-18
> **Rebuttal for Reviewer 2kex. Part [3]**
>
> ### **3. Orthogonality and complementarity with quantization methods**
> We further emphasize that SAES-SVD and weight quantization methods (e.g., GPTQ) are orthogonal and can be combined:
> * Quantization reduces storage by lowering numerical precision.
> * SAES-SVD reduces parameter count and computation via low-rank decomposition.
>
> To validate synergy, we apply 4-bit GPTQ quantization to the factor matrices $U$ and $V$ obtained from SAES-SVD(i.e., $W\approx U\Lambda V$).
>
> As shown in Table **R1-3**, **the combined pipeline (“SAES-SVD + GPTQ”) achieves substantially higher accuracy than standard GPTQ-3bit under equivalent compression.** This demonstrates that SAES-SVD not only serves as an effective standalone technique but also provides a powerful complementary approach for achieving higher compression ratios and improved hardware efficiency.
>
>
> **Table R1-3: Performance of equivalent 3bit which combining SAES-SVD and GPTQ vs Pure GPTQ-3bit on Wikitext2 perplexity and average accuracy of 7 zero-Shot tasks for LLaMA2-7B and Qwen2.5-7B**
> |Model        |Method                 |Wiki PPL|AvgAcc@7|
> |-------------|------------------------|--------|--------|
> |**LLaMA2-7B**|GPTQ (3bit)             |10.17   |0.39    |
> |             |SAES+GPTQ (equal 3bit)  |**7.64**    |**0.44**    |
> |**Qwen2.5-7B**|GPTQ (3bit)            |12.29   |0.51    |
> |             |SAES+GPTQ (equal 3bit)  |**9.87**    |**0.57**    |
>
>
> ---
> ---
> > Q3. Practical integration:
>
>
> We thank the reviewer for raising the question of practical deployability. We have already integrated SAES-SVD into HuggingFace Transformers and vLLM with only minimal changes, successfully reproducing all results reported in the paper.
>
> **1. Integration with mainstream inference frameworks.** SAES-SVD is a low-rank compression framework for linear layers: it only replaces the weight matrices of linear layers with their low-rank factors and does not modify any public interfaces. We have integrated SAES-SVD into our current codebase based on the widely used Transformers and vLLM inference frameworks. Model loading, KV-cache management, tensor parallelism, pipeline parallelism, and other components remain unchanged; we only substitute the weights of Linear layers at load time with the compressed SAES-SVD factors. All accuracy evaluations in the manuscript are conducted using Transformers together with the lm_eval_hardness suite, while the throughput and latency evaluations are performed on vLLM.
>
> **2. Required changes to existing serving infrastructure.** SAES-SVD changes only the parameterization of linear transformations; it does not alter the model architecture or the operator types in the forward pass, nor does it require any modifications to the scheduler, KV cache, attention kernels, or other system components. The existing tensor parallelism / pipeline parallelism mechanisms can also be directly reused. So the required changes to different inference stacks are minimal:
> * In generic PyTorch/HF inference code, the simplest approach is to replace each Linear(W) with two chained linear layers, Linear(A) followed by Linear(B). This is purely a module-level substitution.
> * In highly optimized engines such as vLLM, we implement a fused Triton kernel for low-rank inference of the form
> A(BX): for each tile, the intermediate result of the first matrix multiplication is kept in SRAM and immediately consumed by the second, thereby avoiding redundant activation writes and reads.
>
>
> ---
> ---
> ### **Final Remarks**
>
> We sincerely appreciate your constructive comments. All concerns you raised fall within the scope of improvements achievable through additional experiments and clearer explanations, rather than indicating fundamental issues with the method. In this rebuttal and the revised manuscript, we have systematically addressed all points through:
> 1. **Expanded model coverage:** Additional evaluations on models ranging from 7B to 70B and across architectures such as Qwen, demonstrating strong generalization capability.
> 2. **Modernized and diversified benchmarks:** New experiments on long-context reasoning, mathematical reasoning, and code generation confirm consistent and significant performance gains in challenging real-world scenarios.
> 3. **Broader comparisons and synergistic compression strategies:** We clarify the theoretical orthogonality to quantization methods and empirically validate the benefits of combining SAES-SVD with GPTQ.
>
> **We will fully incorporate all new analyses and results into the revised manuscript. We are deeply grateful for your valuable, constructive, and insightful suggestions—they have greatly enhanced the clarity, completeness, and academic impact of this work.** We believe our rebuttal comprehensively addresses every concern you raised, while also helping to more vividly underscore the unique contributions and core value of SAES-SVD.
>
> Best regards,
> Authors of SAES-SVD

---

> ### Author Response · Authors · 2025-11-25
> **Gentle Follow-up Regarding Pending Reviewer Feedback**
>
> Dear Reviewer 2kex,
>
> **We hope you are doing well. One week has passed since we submitted our full rebuttal, and we would like to kindly follow up to see whether you have had the chance to review our responses.**
>
> In the updated rebuttal, we have substantially expanded the evaluation scope as suggested, including (i) adding long-context, code-generation, and mathematical reasoning benchmarks; (ii) incorporating larger and more diverse architectures such as LLaMA-3.1 and Qwen-2.5; and (iii) providing new cross-paradigm comparisons and analyses. These additions were motivated directly by your comments and significantly improve the comprehensiveness of our study.
>
> If any clarification or further evidence would help your assessment, we would be very happy to respond promptly.
>
> **Thank you again for your thoughtful and constructive feedback.**
>
> Best regards,
>
> Authors of SAES-SVD

---

> > ### Comment · Reviewer_2kex · 2025-11-26
> > **Thanks for your detailed explanation. Looking forward to the release of reproducible GitHub repository**
> >
> > The performance of the proposed approach is impressive, and it demonstrates substantial potential as a robust SOTA method for advancing research in this field. I look forward to the release of the open-source code. All my prior concerns and inquiries have been thoroughly addressed, and I am fully satisfied with the clarifications provided.

---

> > > ### Author Response · Authors · 2025-11-27
> > > **Thank You for Your Positive Assessment**
> > >
> > > Dear Reviewer 2kex,
> > >
> > > Thank you very much for your encouraging follow-up and for confirming that all your concerns have been fully addressed. We sincerely appreciate your thoughtful evaluation and constructive engagement throughout the review process. We are actively preparing the reproducible codebase and will release it publicly as soon as the camera-ready phase permits.
> > >
> > > **Your positive assessment and recognition of the method’s potential are greatly appreciated. Thank you again for your time and valuable feedback.**
> > >
> > > Best regards,
> > > Authors of SAES-SVD

---

### Author Response · Authors · 2025-12-02
**Post-Rebuttal Summary for the Area Chair**

Dear AC

**We sincerely appreciate your time.** In this comment, we would like to provide a concise, structured summary of the discussion so far. In our rebuttal and extended appendix, we have systematically addressed all reviewers’ concerns, and **all reviewers who responded after the rebuttal (2kex and khS9) explicitly confirmed that their concerns are fully resolved.** In particular, Reviewer 2kex **expressed full satisfaction**, describing **SAES-SVD as a strong state-of-the-art method**, and accordingly **raised the overall score.**

Across all four reviews, the comments consistently acknowledge:

* **Clear and novel motivation**: SAES-SVD directly targets cross-layer error accumulation, a concrete pain point of SVD-based LLM compression that prior work largely ignores. Reviewers emphasized that this motivation is strong, well-justified, and clearly articulated.
* **Strong theory and closed-form design**: CEALC + ACES are derived from second-order statistics and retained-energy maximization, yielding an interpretable, training-free low-rank framework. Reviewers noted that the overall method is coherent and principled, and that the analysis provides solid theoretical support for the proposed procedure.
* **Practical effectiveness**: Under a unified, no-finetuning, uniform-rank protocol, SAES-SVD consistently improves perplexity, zero-shot accuracy, and inference speed over strong SVD baselines. Reviewers found the experiments comprehensive and convincing.

The remaining concerns fall into several **extended validation and clarification categories, rather than questioning the core methodology**:

* **broader architectural coverage** beyond LLaMA (e.g., Qwen2.5, LLaMA-3.1);
* **broader task coverage** (e.g., long-context reasoning, code generation);
* **cross-paradigm comparisons** with structured pruning and quantization;
* **compression-time / complexity breakdown** for statistics collection and ACES.

We have conducted a substantial amount of additional work to address these points, which **we summarize reviewer-by-reviewer below.**

---

> ### Author Response · Authors · 2025-12-02
>
> |Reviewer|Strengths|Concerns|Initial rating|Rebuttal actions|Reviewer reaction|
> |:-:|-|-|-|-|-|
> |**2kex**|**❶.Principled second-order foundation with elegant local reconstruction** and weighted cumulative error compensation.**❷.Practical adaptive mechanism:** ACES automatically sets layer-wise weights, removing manual tuning. **❸.Consistent gains over baselines,** notably a much smaller accuracy drop |**❶.Model architectural  diversity:** Need performance of more model architectures(e.g. Qwen). **❷.Task diversity:** Need performance on broader tasks (e.g., long-context reasoning, coding). **❸.Broader baselines:** Seek a wider set of compression methods, including PTQ approaches such as GPTQ. **❹.Practical integration:** Asks for details on integration with popular inference frameworks.| 4|❶Expanding architectural coverage to LLaMA-3.1 and Qwen-2.5, where SAES-SVD **consistently outperforms all SVD baselines**; ❷Adding long-context, coding, and math benchmarks, on which SAES-SVD **consistently narrows the FP gap than others**; ❸Including cross-paradigm baselines and PTQ integration with GPTQ, showing clear gains at iso-compression; ❹Providing concrete vLLM integration details.| Reviewer explicitly stated that **"all concerns have been completely resolved"**, described SAES-SVD as having **“substantial potential as a robust SOTA method”** and **raised the overall score**.|
> | **5MPy** | **❶.Clear, empirically supported motivation:** directly addresses cumulative cross-layer SVD error in LLMs. **❷.Principled design with solid theory:** CEALC and ACES are cleanly formulated and coherently analyzed. **❸.Strong, broad empirical validation:** consistent accuracy/latency gains across models and datasets. | **❶.Robustness analysis:** Requests deeper analysis of FS-FOA and a clearer link between RER and PPL. **❷.Model architectural  diversity:** Seeks broader model-architecture coverage (e.g., Qwen). **❸.Mixed-rank compatibility:** Asks whether SAES-SVD can be combined with mixed-rank schemes. **❹.Relation to AA-SVD:** Requests clarification on differences from the concurrent AA-SVD submission. |4| ❶Deepening the robustness study of FS-FOA and explicitly validating the chain **RER  →...→ lower PPL**; ❷Expanding architectural coverage to **Qwen-2.5**. ❸Showing **mixed-rank compatibility** by inserting SAES-SVD into ASVD-style schedules with **uniform gains at the same compression ratios**; and ❹Clarifying the **conceptual and empirical differences to AA-SVD**, emphasizing our explicit modeling/suppression of accumulated error via RER and the resulting **more stable, superior performance.**|**No response.** Noting the original review explicitly stated **they would "raise the score if these concerns were resolved,"** which we believe we have fully achieved with a multi-part, point-by-point rebuttal.|
> |**khS9**|**❶.Comprehensive experiments with compelling results:** consistently surpasses strong SVD baselines, including fine-tuned ones. **❷.Systematic, interpretable, and efficient:** theoretically sound handling of error accumulation with ACES’s elegant closed-form adaptive tuning.| **❶.Compression-time/complexity breakdown:** Need these statistics vs baseline. **❷.Broader baselines:** Seek comparisons beyond SVD (e.g., pruning) **❸.Combintation with PTQ:** Ask about combining with quantization methods such as GPTQ.|6|❶Adding a detailed **compression-time and complexity breakdown**, showing ours impressive time advantage ❷Introducing **broader cross-paradigm baselines** beyond SVD, including structured pruning methods and quantization, where SAES-SVD is consistently better; and ❸Explicitly **combining SAES-SVD with GPTQ**, demonstrating that **SAES-SVD+GPTQ** outperforms pure GPTQ-3bit, confirming the practical compatibility.|Reviewer explicitly **replied that our answers “resolved my confusion”** and expressed gratitude; |
> |**kM1J**|**❶.Well-motivated and novel:** SAES-SVD tackles a fundamental, previously overlooked limitation of SVD-based compression. **❷.exceptional Mathematically rigorous:** comprehensive CEALC and ACES derivations turn empirical insight into a theoretically grounded advance. **❸.Strong training-free performance:** achieves competitive results with a simple, straightforward design.| **❶.Model architectural diversity:** Seek broader model-architecture coverage. **2.Broader baselines:** Asked for comparisons with structured pruning and quantization **❸.Pointed out the typos** between Figure 3 and the text for Dip-SVD on LLaMA-30B.|6|❶Expanding architectural coverage beyond LLaMA to **Qwen-2.5 and LLaMA-3.1**, where SAES-SVD **consistently better**; ❷Adding **cross-paradigm baselines** with structured pruning and quantization, **showing  SAES-SVD's superiority;** and ❸Correcting the **Dip-SVD typo/mismatch**.|**No Response** We believe we have fully addressed all concerns  with a point-by-point rebuttal.|
>
> We are sincerely grateful for your time, and efforts.
>
> **Best regards,**
>
> **Authors of SAES-SVD**

---

### Meta-Review · Area_Chair_72Af · 2026-01-11

**Summary:**

Main concerns that have been raise by more than one reviewer are about broader architectural coverage beyond LLaMA, broader task coverage, cross-paradigm comparisons with structured pruning and quantization, and compression-time / complexity breakdown. After read the discussion carefully, I think the authors provided suffient results for these concerns.

**Reviewer Concerns:**

Reviewer 5MPy's concern about Theoretical limitations and missing robustness analyses is till outstanding.

**Reviewer Scores:**

Reviewer 2kex confirmed that they are satisfied with the work but did not raise the score.
Reviewer 5MPy's concerns are fully addressed but they did not participate the discussion.

---

### Decision · Program_Chairs · 2026-01-26

Accept (Poster)